# Modeling tissue-specific breakpoint proximity of structural variations from whole-genomes to identify cancer drivers

Alexander Martinez-Fundichely [1,2,3,4] ✉, Austin Dixon [2,5] & Ekta Khurana [1,2,3,4] ✉

Structural variations (SVs) in cancer cells often impact large genomic regions with functional consequences. However, identification of SVs under positive selection is a challenging task because little is known about the genomic features related to the background breakpoint distribution in different cancers. We report a method that uses a generalized additive model to investigate the breakpoint proximity curves from 2,382 whole-genomes of 32 cancer types. We find that a multivariate model, which includes linear and nonlinear partial contributions of various tissue-specific features and their interaction terms, can explain up to 57% of the observed deviance of breakpoint proximity. In particular, three-dimensional genomic features such as topologically associating domains (TADs), TAD-boundaries and their interaction with other features show significant contributions. The model is validated by identification of known cancer genes and revealed putative drivers in cancers different than those with previous evidence of positive selection.

Whole-genome sequencing of cancer genomes has revealed that they contain a wide variety of DNA structural variations (SVs) that include deletions, duplications, translocations, and other complex events[1]. The SVs in cancer cells arise from different mechanisms and vary in size from kilobases to whole chromosomal rearrangements[1–4]. Consequently, SVs usually span several genes and their associated regulatory elements. While it is well known that genomic rearrangements and copy number variations (CNVs) can lead to dysregulation of tumor-suppressors or oncogenes and act as drivers of cancer progression[1,5–8], identification of SVs under positive selection in cancer remains a challenging task. This is because SVs are heterogeneously distributed across the genome leading to many genomic regions recurrently altered in multiple samples due to neutral background processes[4]. To identify the events under positive selection, the null background distribution of SV breakpoints must be characterized by accounting for the genomic covariates[5,9]. Additionally, the identification of the specific functional element that is the target of positive selection (i.e., the

coding sequence of a gene, its *cis*-regulatory regions, or noncoding RNAs) constitutes another challenge due to the large genomic span of SVs.

While numerous computational methods have been developed to model the background distribution of single-nucleotide variants (SNVs) and identify drivers in a tissue-specific manner, similar methods for SVs are lacking[5,10–13]. The Pan-Cancer Analysis of Whole Genomes (PCAWG) SV Working Group used a Gamma-Poisson fit to model the breakpoint density from 2658 whole genomes using eight covariates to identify the significant driver genes in several cancers[5]. However, this analysis was performed at the pan-cancer level without accounting for tissue-specific covariates. Since there is ample evidence of different SV distributions and putative mechanisms across cancer types[1,2,14], it is critical to model SV breakpoint distribution in a tissue-specific manner to obtain the corresponding accurate null models. Importantly, the three-dimensional (3D) high-order genomic structure, such as the topologically associating domains (TADs), has not previously been

[1]Sandra and Edward Meyer Cancer Center, Weill Cornell Medicine, New York, NY 10021, USA. [2]Institute for Computational Biomedicine, Weill Cornell Medicine, New York, NY 10021, USA. [3]Department of Physiology and Biophysics, Weill Cornell Medicine, New York, NY 10065, USA. [4]Englander Institute for Precision Medicine, Weill Cornell Medicine, New York, NY 10021, USA. [5]Present address: Children's National Hospital, Washington, DC 20010, USA. ✉e-mail: alm2069@med.cornell.edu; ekk2003@med.cornell.edu

considered a covariate of breakpoint distribution. Recent studies have analyzed the relationship between 3D topology and SVs in cancer[8,15–17]. It was reported that while SVs can lead to changes in chromatin folding, only 14% of TAD-boundary deletions are associated with significant gene expression changes[16]. This finding indicates that many of these SV events may be related to neutral evolution in tumor cells and must be accounted for in the null model to identify true drivers. Furthermore, previous studies did not account for the nonlinear contribution of covariates or their interactions in the modeling of SV breakpoint distribution.

Here, we describe a method that models the breakpoint proximity of SVs in 2382 cancer whole genomes in a tissue-specific manner. We used nine genomic covariates, including the recurrence of TADs and TAD boundaries across cell lines and tissues, as well as their functional classification based on chromatin states. We implemented a generalized additive model (GAM) to describe the genome-wide SV breakpoint proximity. The use of a GAM allows us to include both the linear and nonlinear contributions of variables as well as their interplay. Modeling breakpoint proximity has the advantage of allowing us to analyze the clustering of breakpoints over dynamic genomic lengths to capture different breakpoint trends, unlike breakpoint-density-based approaches that rely on fixed genomic bin sizes. Finally, we use this approach to identify loci that exhibit signals of positive selection and the functional elements that are the likely targets of selection. The method is implemented in Cancer Structural Variation Drivers (CSVDriver), a user-friendly tool that can be used by researchers to model the SVs from whole-genome sequencing data and identify putative cancer drivers.

## Results

We analyzed a set of 324,838 high-confidence somatic SVs derived from whole-genome sequencing of 2382 patients of 32 cancer types from 15 organ systems (Supplementary Data 1). The cancer types include those with a high SV burden from the PCAWG project[1,18] and metastatic prostate cancer samples[19,20]. Based on the rationale that tissue-specific covariates can influence the rearrangement landscape, the cancer types from different organ systems were modeled separately. Furthermore, the prostatic primary and metastatic cohorts were analyzed separately since they can have distinct drivers.

### Breakpoint proximity curve to model genome-wide SV distribution

To describe the genome-wide distribution of SV breakpoints in a given cohort, we included all breakpoint coordinates for each sample. Then we computed the breakpoint proximity curve (BPpc) based upon the breakpoint neighbor reachability ($BPnr_i$) for each individual breakpoint ($BP_i$) in the cohort. This metric captures the genomic regions with high or low proximity between breakpoints (Methods). The BPpc is defined as the smooth curve resulting from the nonparametric local polynomial regression (locally estimated scatterplot smoothing, LOESS) fitted to the dataset of $BPnr_i$, after reverse scale normalization $-log10(BPnr_i + 1)$ (Fig. 1a). BPpc allows the use of a peak-calling approach to directly identify the regions with higher breakpoint clustering relative to the surrounding area (i.e., peak summits), thereby overcoming the inherent issues associated with predefining a genomic bin size for computing breakpoint density along the genome[5,21]. This is important because functionally relevant breakpoint clustering events may occur over a wide range of genomic lengths. Thus, BPpc models the underlying distribution of breakpoints in a more robust and unsupervised manner compared to the computation of breakpoint densities.

### Expected BPpc using genomic covariates in a GAM

The core of the genomic breakpoint distribution arises under neutral selection due to background processes likely corresponding to the tissue-specific functional and structural genome heterogeneity. We modeled the expected background BPpc per tissue type using a GAM with multiple variables (Supplementary Data 2), including the tissue-specific chromatin state annotations from the Roadmap Epigenomics project[22,23], recurrence of TADs (TAD.recurr), and TAD boundaries (TAD-B.recurr) from the 3D Genome Browser[24], replication timing (RT) from ENCODE Repli-seq data[25] and fragile sites (FS) from the HumCFS database[26]. Other variables include the genome complexity from UCSC genome browser repeat elements data[27], gene density, and GC content (GC). Thus, the expected breakpoint proximity curve is defined as $\widetilde{BPpc} = \beta X + \varepsilon$, a variation of the generalized linear model in which X is a matrix of covariates (breakpoint predictors) that can contribute linearly as well as nonlinearly to the response variable $\widetilde{BPpc}$ (Methods) (Fig. 1b).

The multivariate model explained a larger proportion of the null deviance compared to the single predictor models, even without including the interplay of covariates (Fig. 2a). The median explained deviance is 18% and ranges from 6% for lung cancer to 50% for lymph-nodes (Fig. 2a). We expected that modeling the interaction effects of genomic features may contribute substantially to the model due to the underlying complexity of SV breakpoints in cancer[21]. Since all possible combinations of features make the model computationally infeasible and intractable for a meaningful interpretation, only the interaction terms of features that contribute at least 10% in at least one cohort in single predictor models were included (Fig. 2a). The resulting model accounts for the interaction of various features, including RT, GC, TAD.recurr, TAD-B.recurr and gene density, with lamina-associated domains (LADs) that correspond to the higher-order genome disposition within the nucleus[28]. The model also accounts for the pairwise interplay between gene density and RT, gene density and TAD.recurr, as well as RT and TAD.recurr; in each case, the interactions are modeled separately for distinct TAD-segment classes. The TAD segments are defined based on differential conservation across cell types and annotated into three classes (quiescent, low-active, or active) using enrichment of tissue-specific chromatin annotations[16] (Methods). The multivariate GAM that accounts for the interaction of covariates showed further substantial improvement over the model without the interaction terms. The median explained deviance is 29%, ranging from 10% in lung cancer to 57% in lymph-nodes (Fig. 2a). The model performs well in all cohorts (Fig. 2c) with fairly narrow and symmetric residual distributions (Supplementary Fig. 1). The difference in explained deviance between cancer types could be due to either "missing covariates" or "missing values of covariates" for certain cohorts. However, the difference in the "missing values" in the covariate data availability for each tissue type does not explain the variability in the performance of the model (Supplementary Data 2). We checked if the cell type heterogeneity[29] is related to the explained deviance, and we found a positive correlation (Spearman correlation coefficient = 0.4) though it is not statistically significant likely due to the low number of cancer types ($n = 12$) (Supplementary Data 3 and Supplementary Fig. 2a). Similarly, we do not find a statistically significant correlation between the average number of SVs per donor and explained deviance although the correlation coefficient is −0.1 (Supplementary Fig. 2b). Thus, while we can not check the relationship of multiple features to the explained deviance due to the small number of cancer types, it is likely that signatures of structural variation[1], the evolutionary history and the clonal status of tumors[30,31] may also play a role in the observed variability of the explained deviance for each cohort. Notably, we find a substantial decrease in the correlation of observed vs. predicted BPpc when using the covariates from unmatched tissues (Fig. 2b). This clearly highlights the importance of using tissue-specific covariates.

Input: Structural variation from tumor whole-genome sequencing

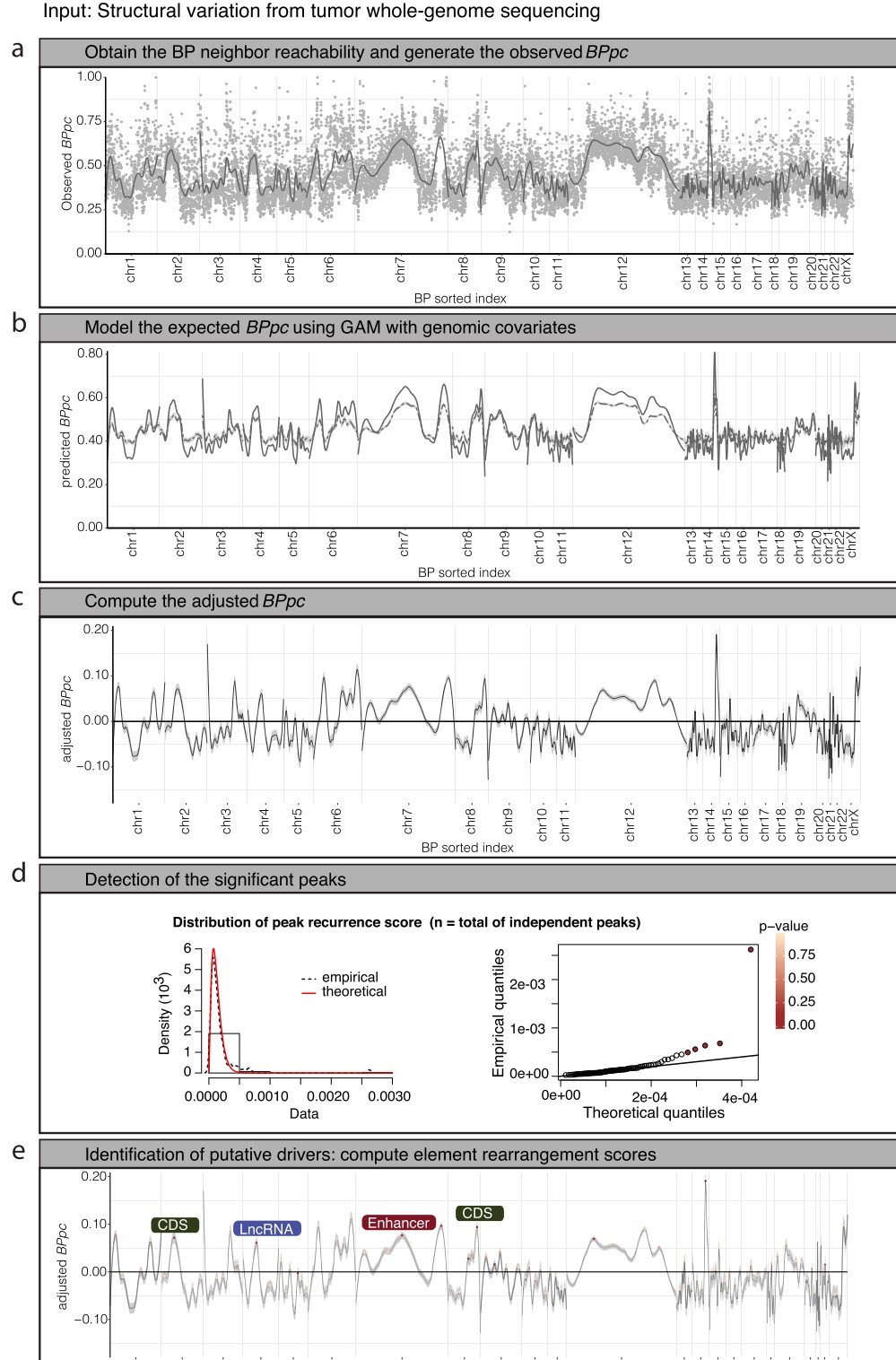

**Fig. 1 | CSVDriver workflow.** The input data are the standard cancer somatic SV calls that include a pair of breakpoint coordinates (BP1, BP2) for each SV_id per sample. **a** Step 1 computes the breakpoint neighbor reachability (BPnri gray dots) for each breakpoint ($BP_i$), where '$i$' is the sorted index. Then, for each chromosome, the $BPpc$ is generated as the smooth curve (gray line) resulting from the non-parametric local polynomial regression fitted to the dataset of $BPnr_i$, after reverse scale normalization $-log10(BPnr_i + 1)$. **b** In step 2, based on the observed $BPpc$ distribution (solid line), the method assesses the expected ($\widetilde{BPpc}$) background distribution (dashed line) by using a generalized additive model (GAM) that includes multiple tissue-specific breakpoint covariates. **c** In step 3, it computes adjusted $BPpc$ (observed − expected). **d** In step 4, the method calls peaks across the adjusted $BPpc$ and out of the total number of independent peaks (n) the method identifies those that potentially correspond to positively selected loci. It computes the peak recurrence score ($PRs$) and based on the empirical density of $PRs$ (dashed line distribution) using a test of fit for the Gamma distribution, it identifies the peaks with $PRs$ significantly higher (QQ-plot FDR < 0.2) than the fitted theoretical density (red line distribution). **e** In the last step 5, the driver candidates are identified as the genomic elements (CDS (coding sequence), enhancer, CTCF-Insulator, and lncRNA) with the highest rearrangement scores within the peak region. The gray shade around the curves (loess smoothing) displays the 95% confidence interval.

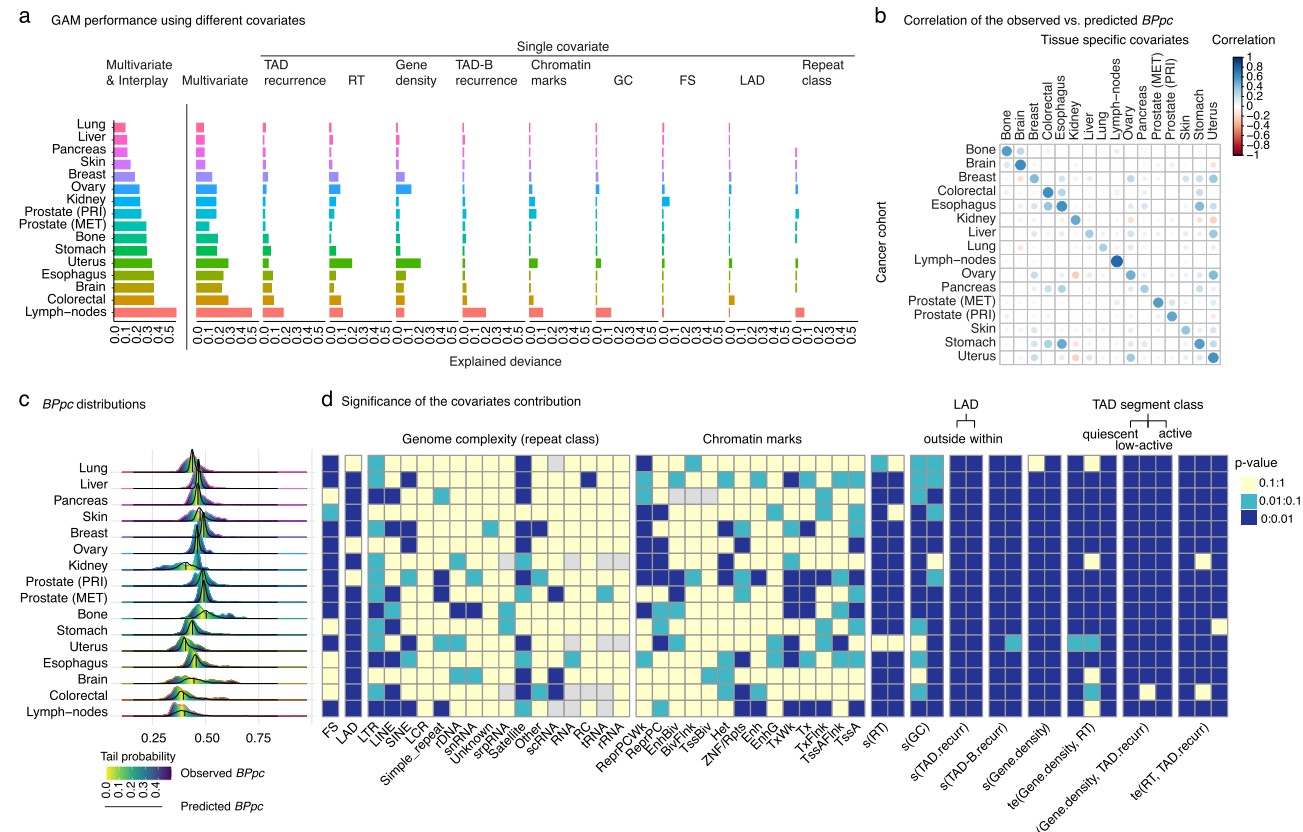

**Fig. 2 | Performance of the GAM for the expected background BPpc for each cancer type. a** Explained deviance of the GAM for the multivariate model with and without interplay and single covariate models. **b** Correlation plot of the performance of the model for each cohort using different tissue-specific covariates. **c** Observed BPpc distributions and heatmap of the tail probabilities. Black curve is the predicted BPpc distribution from the GAM model. **d** Heatmap of the significance of the partial contribution of each covariate to the explained deviance of the model with respect to the mean of the distribution (null intercept). The p-values are calculated by using Bayesian estimated covariance matrix of the parameter estimators. FS (fragile sites), LAD (lamina-associated domain), DNA sequence repeat classes that include LTR (long terminal repeat elements), LINE (long interspersed nuclear elements), SINE (short interspersed nuclear elements) including ALUs, LCR (low complexity repeats), simple repeats i.e., micro-satellites, DNA repeat elements (rDNA), RNA repeats (including tRNA, rRNA, snRNA, scRNA, and srpRNA), satellite repeats, other repeats including class RC (Rolling Circle), and unknown complex sequence. ChromHMM tissue-specific chromatin marks that include TssA (Active TSS); TssAFlnk (Flanking Active TSS); TxFlnk (transcription at gene 5′ and 3′); Tx (Strong transcription); TxWk (Weak transcription); EnhG (Genic enhancers); Enh (Enhancers); ZNF/Rpts (ZNF genes and repeats); Het (Heterochromatin); TssBiv (Bivalent/Poised TSS); BivFlnk (Flanking Bivalent TSS/Enh); EnhBiv (Bivalent Enhancer); ReprPC (Repressed PolyComb); ReprPCWk (Weak Repressed PolyComb). The 's' is a thin plate regression spline smooth function that describes the nonlinearity in the contribution of the replication timing (RT), GC content (GC), gene density (gene.density), and the recurrence of the topologically associated domain (TAD.recurr) as well as the recurrence of TAD-boundary regions (TAD-B.recurr). The partial contribution of each 's' function accounts for the interaction with the corresponding status of LAD. The 'te' is a tensor product interaction that describes the interplay between gene.density vs. TAD.recurr, gene.density vs. RT, and TAD.recurr vs. RT. The partial contribution of each 'te' interplay accounts for the interaction with the corresponding class of TAD segment that includes quiescent, low-active, and active regions. Source data are provided as a Source Data file.

## Contribution of genomic covariates to the background *BPpc* distribution, nonlinearity, and covariates interplay

We find that the contribution of the chromatin state annotations and repeat elements is limited, while the 3D-genome features of TAD recurrence and TAD-B recurrence contribute significantly to the model for most cancer types (Fig. 2d). This result is consistent for both the genomic loci close to the membrane of the nucleus, i.e., -within LAD-, as well as the inner-nucleus genomic regions, i.e., -outside LAD- (Fig. 2d). This suggests that the 3D chromosomal conformation plays an important role in the expectation of the genome-wide distribution of SVs. The results also show that fragile sites, replication timing, GC content, and gene density contribute significantly to many cancer types, although the behavior can vary within vs. outside of LADs for some cohorts (Fig. 2d). The two-dimensional (2D) contour plots allow us to visualize the partial contributions of features to *BPpc*. Most cohorts exhibit peaks and valleys on the smooth function for the partial effects of most covariates, demonstrating the importance of nonlinear modeling

(Fig. 3a, b, Supplementary Fig. 3). For instance, the partial contribution of TAD recurrence shows nonlinear behavior in the brain, colorectal, kidney, and prostate cancers although it is linear in breast cancer regardless of the LAD status (Fig. 3a). The behavior of some features may also vary within vs. outside LADs. For example, the contribution of GC content in breast cancer is linear inside LADs but nonlinear outside LADs (Fig. 3b).

We find a statistically significant contribution for the interaction terms of covariates across all the three functional classes of TAD segments for most cohorts (Fig. 2d). However, their effect sizes show distinct behavior for each cancer type and often across TAD-segment classes as evident from the 2D contour plots (Fig. 3c, Supplementary Fig. 3). Interestingly, the distinct behavior is also apparent for primary vs. metastatic prostate cancers, likely pointing to different processes contributing to early vs. late genome-wide SV distribution[32,33]. In general, there is wide variability in the partial contributions of different features across cancer types, demonstrating the importance of tissue-specific modeling.

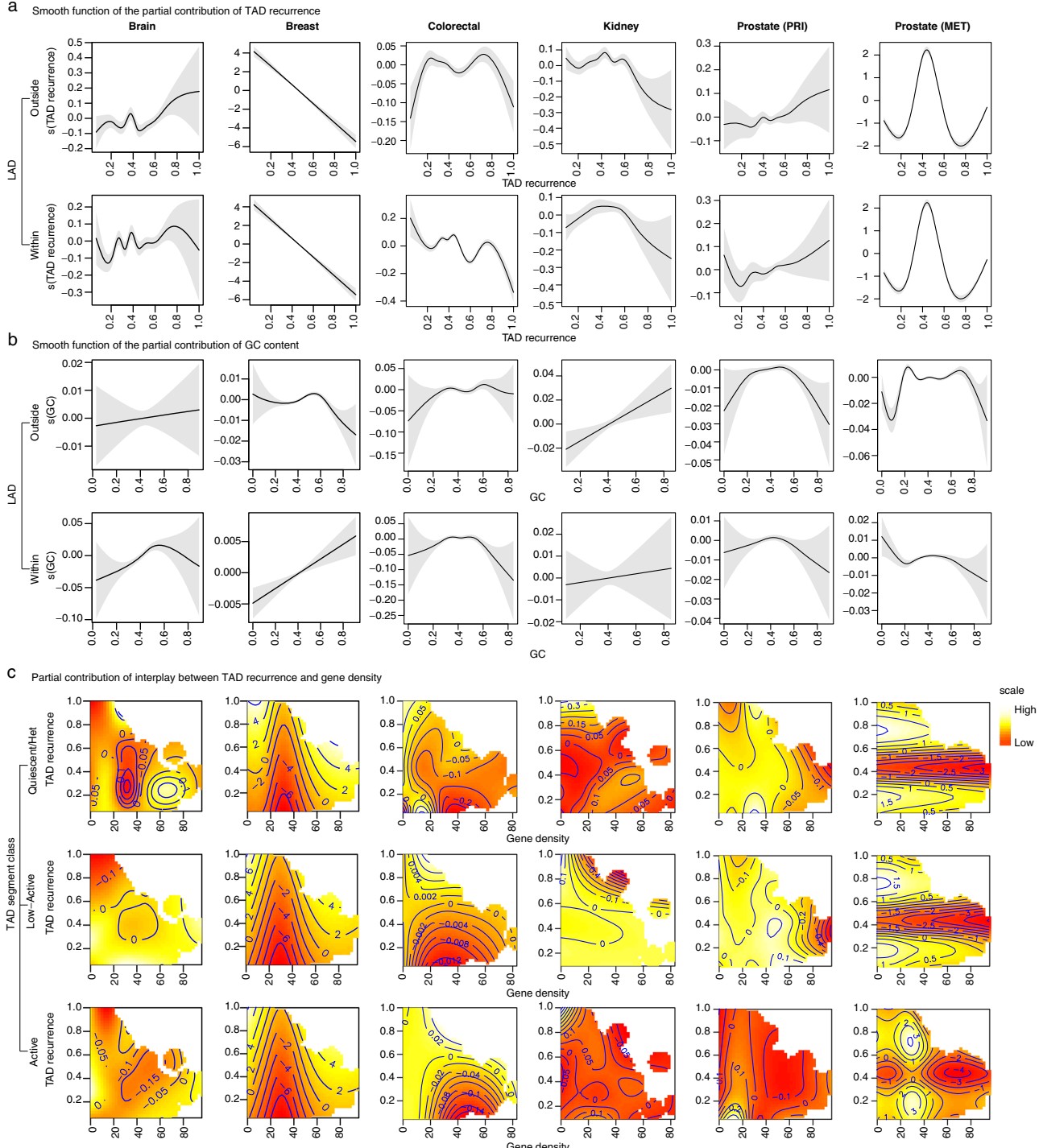

**Fig. 3 | Graphics to analyze the results of the partial contribution of covariates to the GAM shown for representative cancer types. a** The smooth function of partial contribution of TAD recurrence. **b** The smooth function of the partial contribution of GC content. These graphics represent the linear or nonlinear behavior of the partial contribution of each covariate analyzed. The x-axis is the value of the covariate and the y-axis is the corresponding partial effect of the covariate. The gray shade area displays two standard errors above and below the GAM estimate of the smooth curve. **c** 2D graphics for the partial contribution of the interplay between gene density and TAD recurrence for the interaction with the three functional classes of TAD segments. The scale from light yellow to red represents partial contribution for higher to lower values of the distribution. The full set of graphics for all cohorts is shown in Supplementary Fig. 3.

## Detecting significantly recurrent peaks in *BPpc*

We obtained the adjusted $\dot{BPpc} = BPpc - \widetilde{BPpc}$ by correcting the observed curve with the expected model (Fig. 1c). Values around zero in the adjusted curve correspond to the observed *BPpc* close to the expected one from the GAM. Peaks in positive values of $\dot{BPpc}$ correspond to regions where the breakpoints are closer than expected, while the valleys in negative values are loci where the breakpoints are sparser than expected (Fig. 4a and Supplementary Fig. 4). As expected,

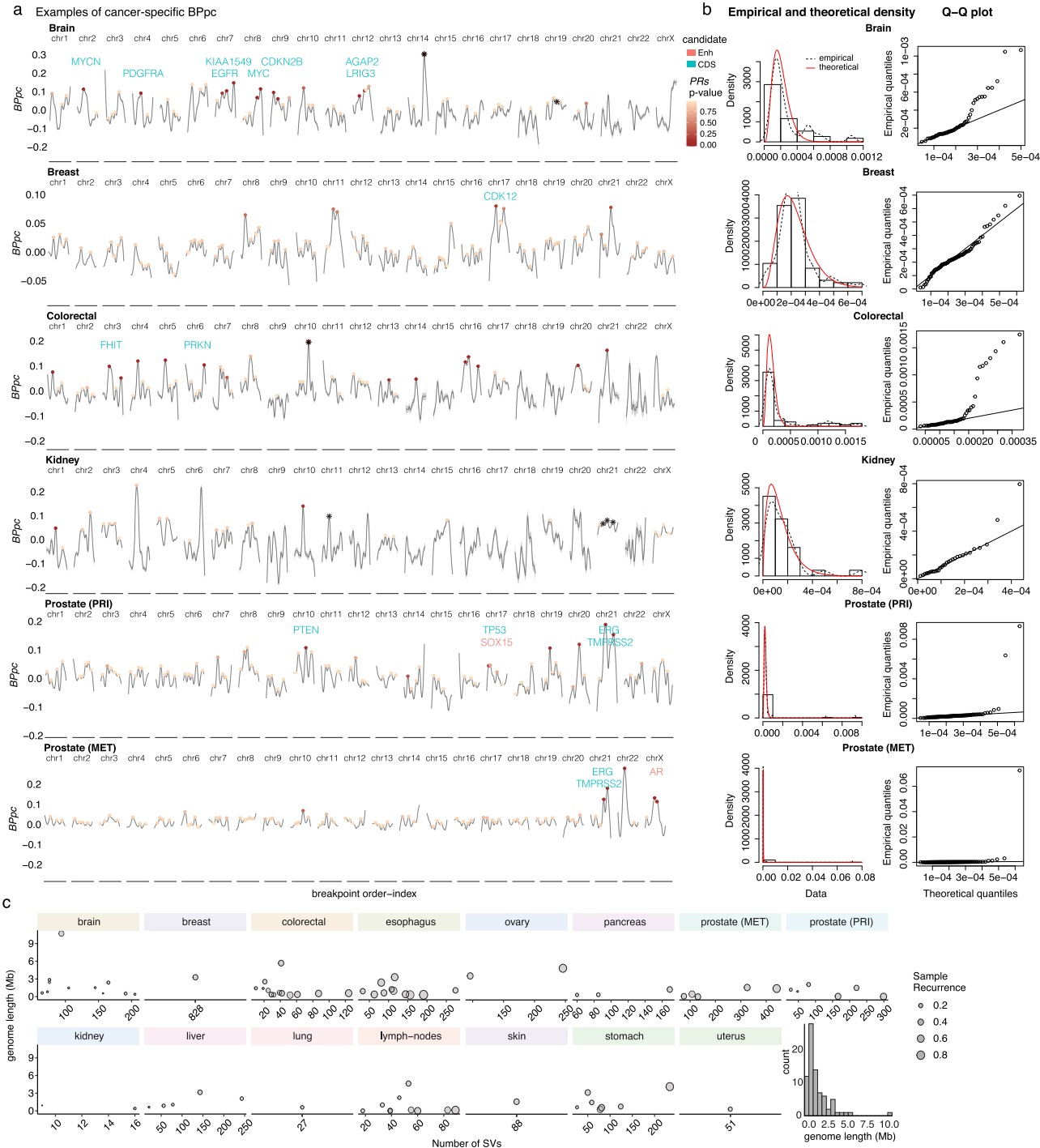

**Fig. 4 | Results for the significantly recurrent rearrangement peaks. a** Examples of cancer-specific *BPpc* and the peaks of significantly recurrent rearrangements for representative cancer types. Each peak shows a dot colored in the scale of significance for the corresponding peak recurrence score (*PRs*). The peaks that come mostly from a unique sample are marked with an asterisk. The known driver candidates detected within the significantly rearranged peaks are marked (green for coding sequence and red for enhancers). **b** Empirical and theoretical density of *PRs* for each cohort and the corresponding QQ-plots which show the *p*-values using a test of fit for the Gamma distribution. The full set of graphics for all cohorts is shown in Supplementary Figs. 4, 5. **c** Scatter plots of the genome length of the peak region vs. number of SVs for each peak, the size of the dots shows the recurrence in the cohort. Histogram of the genome length distribution of the significant peaks. Source data are provided as a Source Data file.

there is a wide variability of the peaks and valleys representing the differential landscape of rearrangements for each cohort (Fig. 4a and Supplementary Fig. 4)

To identify the peaks that potentially correspond to positively selected loci across the *BPpc* rearrangement landscape, we computed

the peak recurrence score (*PRs*) (Methods).

$$PRs = \frac{N_{smp}}{N_{SV}} \times \frac{\text{Peak}_A}{\text{Peak}_{GR}} \qquad (1)$$

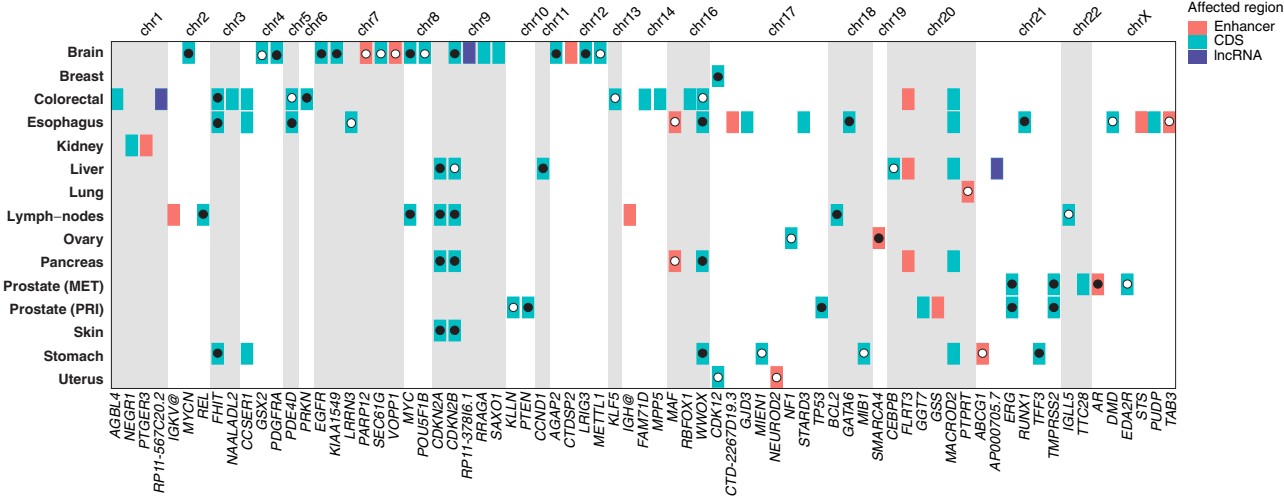

**Fig. 5 | Driver candidates predicted for each cancer cohort.** In green are the genes affected in the coding sequence and in red are the genes impacted at their enhancers. Genes marked with a black dot are the ones previously reported in the corresponding cancer type. Genes marked with a white dot are the ones previously reported as cancer genes but in a different cancer type than our prediction. The IGH@ locus contains four enhancers. The entire list of candidates is shown in Supplementary Data 5.

where $Nsmp$ is the number of unique samples in the peak, $Nsv$ is the number of unique SVs, $Peak_A$ is the area under the peak, and $Peak_{GR}$ is the genomic range of the peak. Thus, $PRs$ is the highest for peaks where many samples (high $Nsmp$) contribute the same SVs (low $Nsv$) to create tight clusters of breakpoints (high $Peak_A$) over narrow regions (low $Peak_{GR}$). Next, for each cohort, we identified peaks with $PRs$ significantly higher than the fitted theoretical density using a Gamma distribution (FDR < 0.2) (Fig. 1d, Fig. 4b and Supplementary Fig. 5).

We identify 79 significantly recurrent peaks that potentially correspond to positively selected loci (Supplementary Data 4). We find that cohorts with more SVs per sample have more observed total peaks as expected (Supplementary Fig. 2c), but there is a negative correlation between the total number of peaks and the number of significant peaks (Supplementary Fig. 2d). The peak summits corresponding to putative driver candidates range from 179 bp to 10.71 Mb, with a median of 822.96 kb, highlighting the strength of our approach to capture breakpoint clustering over varied genomic lengths (Fig. 4c). The number of significant peaks ranges from 0 in bone cancer to a maximum of 13 in colon cancer (Fig. 4c and Supplementary Fig. 6). Upon comparison with PCAWG results[5], we found that the unique peaks identified as significant in our analysis tend to be more cancer-type-specific. Among the 42 peaks that overlap with regions of PCAWG candidates (Supplementary Data 5), 16 are cancer-type-specific, while 26 were found in multiple cancer types. Conversely, 32 peaks have no overlap with PCAWG results: 25 of those peaks are cancer-type-specific and 7 peaks were found in multiple cohorts. Thus, there is a significant enrichment of cancer-type-specific candidate peaks identified by our approach relative to PCAWG (chi-squared test $p$-value = 0.0006), which further demonstrates the power of tissue-specific analysis.

We further investigated the SV type composition of each significant peak for each cohort (Supplementary Data 6). We observed heterogeneity in the SV types within the peaks, reinforcing that a single driver candidate can be affected by different SV types and mechanisms. Nevertheless, we also found regions significantly enriched in one particular SV type compared to other regions (Supplementary Data 6).

## Putative driver candidates identified at significantly recurrent peaks in *BPpc*

We proceeded by predicting the functional elements that are the most likely targets of positive selection within the significantly recurrent peaks by computing the element rearrangement scores ($RS_E$) for all

protein-coding exons, long noncoding RNAs (lncRNAs), enhancers, and CTCF-insulators in the 79 significantly recurrent peaks (Fig. 1d, e, Supplementary Fig. 6 and Supplementary Data 7) (Methods). The highest $RS_E$ values point to elements impacted by the largest number of SVs in the maximum number of samples within the entire peak. Across all the analyzed cancer cohorts, we identified 53 coding genes, 24 enhancers of 17 other genes, and 3 lncRNAs as the most likely targets (Fig. 5) (Methods). Our results are validated by known cancer genes in different cohorts. For example, *TMPRSS2–ERG* fusion[34,35], and *PTEN* and *TP53* deletions[36,37] in prostate cancer. *AR* enhancer is significantly affected only in the metastatic prostate cancer cohort, which is consistent with the development of the disease[36–38]. We also found *EGFR, MYCN*, and *MYC* in brain cancers;[39,40] *BCL2, MYC,* and the loci of *IGH* translocations in lymph-nodes cancer;[41–43] *CDK12* in breast cancer; *CCND1* in liver cancer; and *RUNX1, GATA6,* and *PDE4D* in esophageal cancer (Fig. 5). Besides these known cancer genes in specific cohorts, we also identified several others with known roles in multiple cancers (Fig. 5). Interestingly, most of these common candidates are large cancer genes within fragile sites, such as *FHIT, WWOX, CCSER1, IMMP2L, CDKN2A,* and *CDKN2B*[26,44–47]. While it remains unclear whether the regions of fragile sites have meaningful implications in cancer progression, by accounting for these sites as covariates in our model, our findings reveal the specific cancer types where these fragile site genes are more likely to have tumorigenesis activity via increased genomic instability[48,49].

Overall, out of the 73 genes whose exons or enhancers are identified as putative drivers, 47 are known cancer genes (Fig. 5). Genes identified as potential drivers in our analysis that are known to be oncogenic in another cancer type[50,51] are of high interest since they are often therapeutic targets under investigation (Fig. 6). For example, *DMD* is known to be a cancer gene[50] in human myogenic tumors, such as gastrointestinal stromal tumors[52], rhabdomyosarcoma[53], and leiomyosarcoma[54], and we find it is a putative driver gene in esophageal cancer where 54% of the cohort carries SVs in this region (Fig. 6a). We also identified *LRRN3* as a driver candidate in esophageal cancer with SVs in 20.7% of the samples (Fig. 6b). Other members of the *LRRN* gene family have been found as drivers in multiple cancer types, including neuroblastoma and gastric cancer[55], and the role of *LRRN3*, in particular, has been studied in fibrosarcoma[56]. As shown in Fig. 6, clear peaks in *BPpc* correspond to tight clustering of breakpoints allowing us to confidently pinpoint these genes as putative drivers.

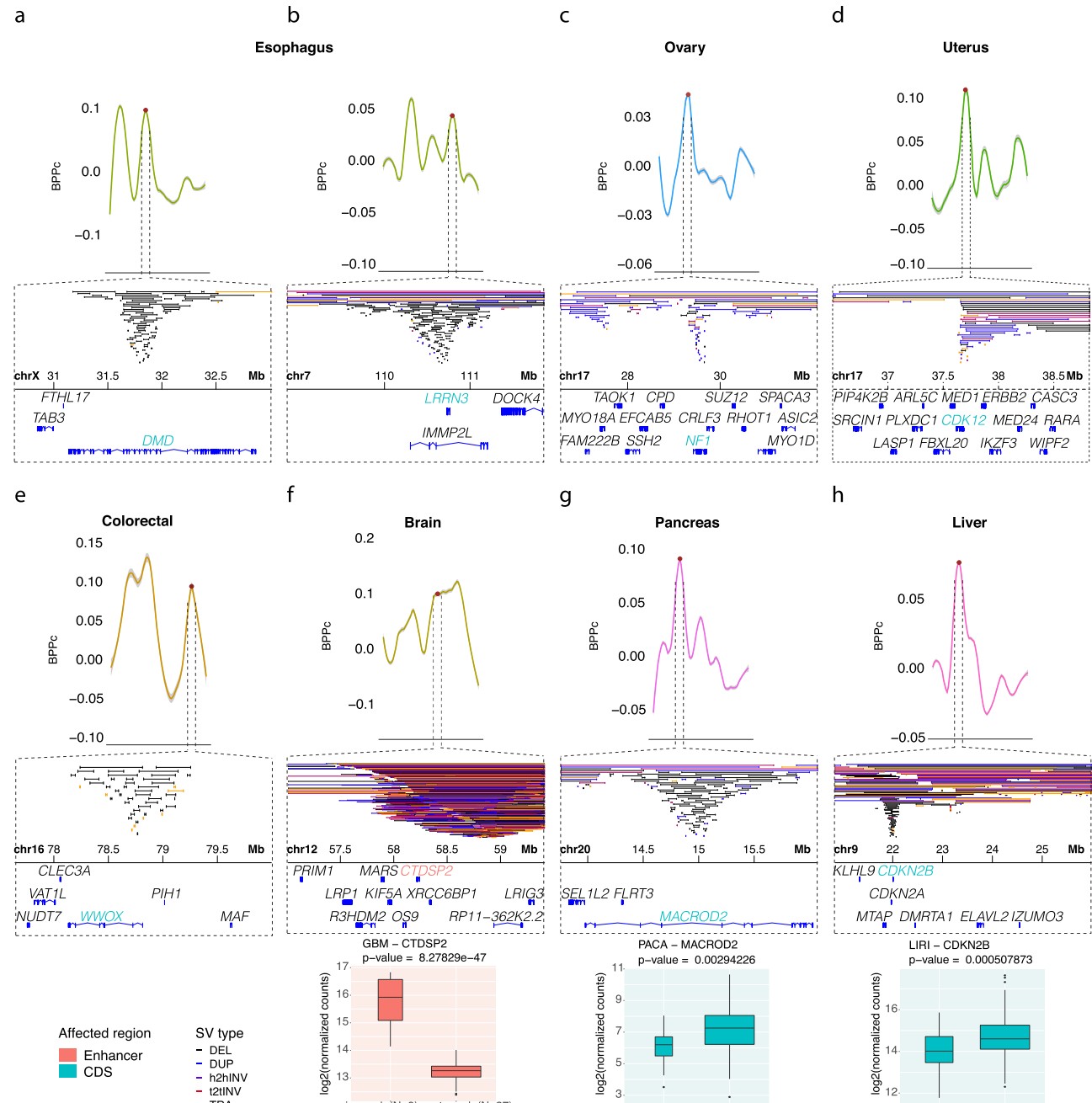

**Fig. 6 | Zoomed in plots of the genomic location of the significantly rearranged peaks that show breakpoint clustering at regions of candidates with previous evidence in other cancer types.** The different types of SVs are color coded as (DEL black, DUP blue, h2hINV purple, t2tINV red, TRA orange). **a** *DMD* in a peak on chrX in esophageal cancer. **b** *LRRN3* in a peak on chr7 in esophageal cancer. **c** *NF1* in a peak on chr17 in ovarian cancer. **d** *CDK12* in a peak on chr17 in uterine cancer. **e** *WWOX* in a peak on chr16 in colorectal cancer. **f** *CTDSP2* in a peak on chr12 in brain cancer. **g** *MACROD2* in a peak on chr20 in pancreatic cancer. **h** *CDKN2B* in a peak on chr9 in liver cancer. The boxplots show the differential expression between samples with and without SVs in the peak region. GBM glioblastoma multiforme, PACA pancreatic cancer, LIRI liver cancer. In green are the genes affected in the coding sequence. In red are the genes impacted in their enhancer regulatory region. The middle hinge corresponds to the median. The lower and upper hinges correspond to the first and third quartiles (the 25th and 75th percentiles). The upper whisker extends from the hinge to the largest value no further than 1.5 × IQR from the hinge, and the lower whisker extends from the hinge to the smallest value at most 1.5 × IQR of the hinge (IQR is the interquartile range, or distance between the first and third quartiles). Data beyond the end of the whiskers are plotted individually as outlier points. We use Wald statistical test (from DESeq2) to compute the *p*-values and Benjamini and Hochberg method to corrected for multiple testing.

Another interesting result is our finding in ovarian cancer of a region affected in 25% of the cohort where our method points to *NF1* as a potential driver (Fig. 6c). Previous studies have shown the importance of *NF1* in several other cancers, including glioblastoma, melanoma, breast, and lung cancers[39,57–59]. The gene *CDK12* is a candidate within a region affected in 14.9% of the uterine cancer cohort (Fig. 6d). *CDK12* is a well-studied target in other female reproductive cancers, such as ovarian and breast[60], and it has also been studied in stomach and prostate cancers[20,61]. In colorectal cancer, we predict *WWOX* as a putative driver in a region that shows SVs in 25% of the samples (Fig. 6e). *WWOX* is within a known fragile site and has been reported to be important in several studies of different tumor types, including breast, prostate, lung, esophagus, cervical, ovarian, and bladder cancers[62–66].

Limited availability of RNA-seq data prevented analysis of the impact on gene expression for many candidates (Supplementary Data 8). Among the cohorts with sufficient sample sizes for RNA-seq analysis, we find that an enhancer of *CTDSP2* is a putative driver in brain cancer (glioblastoma multiforme), and the SVs are associated with its differential expression (Fig. 6f). *CTDSP2* is known to be important in other cancer types[67,68]. The genes *MACROD2* in pancreatic cancer (Fig. 6h) and *CDKN2B* in liver cancer (Fig. 6g) also show significant differential expression between samples with SVs at these loci relative to those without SVs.

These findings highlight the importance of both the tissue-specific and clustering-based approaches used in our method to capture the significantly rearranged regions in different cancers. Notably, while breakpoint clustering clearly points to these specific candidates, as seen in Fig. 6, breakpoint density over fixed bins would require different bin sizes to capture the ones with maximum density. This is because using a fixed-size window for all these instances will likely generate several bins with high density without clearly pointing to the most probable target element of selection.

### Power of detection for the significantly recurrent peaks in *BPpc*

We used a binomial model to analyze the power of detection of significant peaks, defined as the probability to find the expected number of samples for predicting the significant peaks, similar to the approach used for power calculations by PCAWG[5,69]. The results of the power calculations show that although the cohort size for esophagus, stomach, lung, uterus, and breast cancers is borderline, the sample size currently available for most cohorts provides 90% power of detection for peaks with a prevalence of 25% or more (Supplementary Fig. 7). However, the 94 donors in bone cancer provide ~38% detection power for such peaks. We found that for lower frequency events (5% or lower), the detection power ranges from ~10 to ~75%, so the sample size is insufficient for any cohort to reach 90% power of detection. These calculations provide a potential explanation for the different number of significant peaks across cohorts. They show that for events of frequency 5% or lower, we do not have sufficient power for any of the cohorts. While at least 100–200 genomes are needed for most cohorts, others, including prostate and kidney, need more than 250 genomes, whereas the liver, brain, and pancreas need more than 300. The number of genomes available for each cohort is marked in Supplementary Fig. 7.

We found that smaller genomic regions (narrow peaks) need fewer samples to get 90% of the detection power while larger genomic areas (wider peaks) need more samples to reach that power (Supplementary Fig. 7). When the number of samples exceeds the sample size of the current cohorts, some cases show ambiguity in this trend for low prevalence prediction. This is likely due to the uncertainty in the empirical estimate of the breakpoint rate factor and peak's genomic length used in the binomial model, which may change with increased sample size.

### Detecting single-sample peaks in *BPpc*

Besides the main goal of detecting the significant recurrent loci potentially under positive selection, the method annotates the peaks that originate from one or a small number of samples. We compute the single-sample rearrangement score $SSRs$, which evaluates the average contribution of each sample to the total number of breakpoints within each peak (Method). Higher $SSRs$ points to regions originating from fewer samples. We found 12 regions for which the breakpoint load is significantly enriched in fewer samples than expected, in addition to the peaks arising from only one or two samples (Supplementary Data 9). We further compared the features of the significant single-sample peaks with the significantly recurrent peaks predicted to be under positive selection. We do not find any significant difference in the number of SVs, peak area, peak heights, peak genomic range, or

distance to the nearest neighboring peak (Supplementary Fig. 8). We find that single-sample peaks harbor significantly fewer breakpoints than multiple-sample peaks as one might expect.

### CSVDriver: computational tool to identify SV drivers from whole-genome sequences

The computational approach developed in this work to identify SVs and functional elements under positive selection in cancer whole-genomes is implemented in CSVDriver, Cancer Structural Variation Drivers, a user-friendly tool. The input for the tool is SV calls, and researchers can provide the tissue-specific covariates data or use existing datasets to run the tool on their cancer cohort/s. Besides the list of functional elements that are putative drivers in a given cohort, the tool provides graphical visualization of the GAM results, including the analysis of nonlinearity and covariates interaction. To test whether the model can be applied in independent cohorts of the same cancer types, we evaluated three datasets from ICGC cohorts, including breast cancer (549 samples), prostate cancer (396 samples), and skin cancer (206 samples), each of which is independent of the PCAWG datasets used to develop the method. We compared the top significant peaks ($p$-value < 0.025) obtained by adjusting the *BPpc* with the new computed ICGC model (Supplementary Data 10a left), with the top significant peaks ($p$-value < 0.025) obtained using the previously computed PCAWG model on cohorts of the same cancer type (Supplementary Data 10a right). We confirmed that significant cancer-specific loci target the same genomic regions regardless of the model used. Moreover, we find that all candidates identified in PCAWG cohorts are identified in the ICGC cohorts, while three additional candidates are identified in the ICGC cohorts, likely due to the larger sample sizes.

## Discussion

One of the major challenges in cancer genomics is the accurate estimation of the expected heterogeneous distribution of passenger variants. This distribution represents the null background, which can be used to identify loci that exhibit significantly recurrent variants likely due to positive selection. While this problem has received considerable attention for SNVs, studies for tissue-specific neutral models of the genomic distribution of SVs are lacking. We find that a GAM is able to describe the breakpoint proximity distribution of SVs in cancer genomes, with the explained deviance ranging from 10% in lung to 57% in lymph-nodes cancer. The model's explained deviance is affected by the capacity of the chosen set of covariates to capture the patterns of breakpoints and explain the *BPpc* distribution related to background rearrangement events. The variability in the model's performance across cancer types suggests that some missing cancer-specific covariates should be investigated.

The use of a GAM allows us to interpret the results graphically and provides estimates for both the magnitude and statistical significance of the contribution of each feature. We find that the 3D chromosomal conformation plays an important role in the genome-wide distribution of SVs for most cancer types with TAD recurrence, TAD-B recurrence, and their interaction terms with other covariates contributing significantly to the model.

Our method is able to identify the known cancer drivers and further nominate candidates that exhibit higher breakpoint proximity than expected by random chance, such as *DMD* and *LRRN3* in esophageal cancer, *NF1* in ovarian cancer, *CDK12* in uterine cancer, and *WWOX* in colorectal cancer. We note that the pan-cancer analysis by PCAWG allowed the identification of regions that are not likely to gain significance in single cancer analyses due to limited cohort sizes but failed to identify many regions that gain significance in our tissue-specific analysis. While our method is able to identify 24 enhancers as putative drivers, they are usually in the vicinity of other coding exons with similarly high element rearrangement scores, and further

functional validation will be needed to decipher their role in tumorigenesis. However, one prominent example supported by RNA-seq data is *CTDSP2* enhancer, which is a candidate driver in glioblastoma.

Although we could identify the features that contribute significantly to the breakpoint proximity curve, we find that the relationships are usually tissue-specific, complex, and nonlinear, often forbidding straightforward interpretations. Furthermore, while we focused our analysis on the peaks in *BPpc*, the valleys could potentially provide insights about the loci showing depletion of breakpoints due to negative selection in future studies. Finally, larger sample sizes are needed to comprehensively capture all the cancer-specific significantly rearranged regions under positive selection, particularly for events with low prevalence.

## Methods

### Cancer somatic structural variations data

For each cancer type, we use a high-confidence dataset of somatic SV breakpoints. It was obtained from the consensus SV calls of PCAWG Structural Variation Working Group 6[1,18]. We selected the cohorts based on the median number of SVs per sample, aiming to get data representative of the cancer types more impacted by genomic rearrangements. It covers 15 different organ systems, including 32 distinct histological cancer subtypes. They are Skin: MELA (Melanoma), SKCM (Skin Cutaneous melanoma); Pancreas: PACA (Pancreatic Cancer), PAEN (Pancreatic Endocrine Neoplasms); Lymphatic system: DLBC (Lymphoid Neoplasm Diffuse Large B-cell Lymphoma), MALY (Malignant Lymphoma); Esophagus: ESAD (Esophageal Adenocarcinoma); Lung: LUAD (Lung Adenocarcinoma), LUSC (Lung Squamous cell carcinoma); Uterus: UCEC (Uterine Corpus Endometrial Carcinoma); Ovary: OV (Ovarian Cancer); Breast: BRCA (Breast Cancer); Brain (central nervous system): PBCA (Pediatric Brain Cancer), LGG (Brain Lower Grade Glioma), GBM (Brain Glioblastoma Multiforme); Bone: SARC (Sarcoma), BOCA (Bone Cancer); Colorectal: COAD (Colon Adenocarcinoma), READ (Rectum Adenocarcinoma); Kidney: KICH (Kidney Chromophobe), KIRC (Kidney Renal Clear Cell Carcinoma), KIRP (Kidney Renal Papillary Cell Carcinoma), RECA (Renal clear cell carcinoma); Liver: LICA (Liver Cancer), LIRI (Liver Cancer−RIKEN), LIHC (Liver Hepatocellular carcinoma), LINC (Liver Cancer−NCC); Stomach: GACA (Gastric Cancer), STAD (Gastric Adenocarcinoma); Prostate: EOPC (Early Onset Prostate Cancer), PRAD (Prostate Adenocarcinoma). Particularly for prostate cancer, we additionally collected SVs from a cohort of 124 metastatic samples (379,605 SVs) obtained from calls reported in refs. 19 and 20. These datasets provide information from whole-genome sequencing of 2382 cancer donors allowing us to analyze 324,838 high-confidence somatic SVs. We analyzed SVs in autosomes and chromosome X only. The genome reference used is hg19 build. The details are shown in Supplementary Data 1.

### Genomic feature annotations and tissue-specific epigenomic covariates

We use several genome features as the *BP* covariates in the modeling of the expected background of *BPpc*. They include tissue-specific chromatin state marks from Roadmap Epigenomics Mapping Consortium[23], which annotates genomic regions based on histone modifications and chromatin DNA accessibility[70]. We used the average RT signal per 1 Mb genomic bins across eight different cell types (liver, breast, brain, lymphoma system, skin, blood, lung, and prostate) as described in the CNC-Driver method[11]. The datasets were collected from ENCODE and constitute wavelet-smoothed Repli-seq data[25]. The GC content was computed in windows of 101 bp centered in each BP using the function 'GCcontent' from the R Package 'biovizBase' version 1.30.1[71]. The gene density was computed per 1 Mb window using the function kpPlotDensity from the R Package kpPlotDensity version 1.10.0[72]. The chromosomal Fragile Sites (FS) annotation was obtained from the database HumCFS[26]. The information about the genome

complexity was assessed using the genome repeat class (repClass) obtained from repeat masker data in UCSC genome browser[73].

In the set of covariates, we included two higher-order structural 3D chromosomal conformations. The annotation of genome LADs obtained from Akdemir, K. C. et al.[16], and the annotations of genome TADs obtained from the 3D Genome Browser[24], which include a collection of 37 samples (cell lines and tissues) across 19 tissue types (Supplementary Data 2). Although TAD structure tends to be conserved across cell types[74,75], there is evidence that cancer cells show higher TAD structural variability[76,77]. For our tissue-specific analysis, we use the recurrence of TAD and TAD-boundaries (TAD-B) regions. The recurrence of each TAD region was computed as the number of samples (cell lines and tissues) with a minimum of 70% overlap of the TAD regions divided by the total number of samples. TADs with low recurrence (<0.5) point to the regions with high 3D structural variability and potential for tissue specificity, while TADs with high recurrence (>0.5) are regions of similar structure across all datasets. We further classified each sub-segment of TAD that shows distinct recurrence (Supplementary Fig. 9a) into tissue-specific classes of chromatin state by computing the enrichment of 15 tissue-type chromatin marks, similar to the approach used in ref. 16 (Supplementary Fig. 9b). We found that three principal components explain most of the variance in coverage (Supplementary Fig. 10). Consequently, we grouped the TAD segments in three clusters according to their coverage of chromatin marks. The three clusters are quiescent/heterochromatin, low-active, and active chromatin. We confirmed that the clustering is robust across the 16 cohorts analyzed in this study (Supplementary Fig. 11). More details of the sources of genomic feature annotations are shown in Supplementary Data 2.

The putative functional effect for each significantly rearranged locus is predicted by annotating the potential drivers on the basis of the impact of the SVs breakpoints on the coding and noncoding elements within these regions. For the coding drivers, we use the subset of protein-coding genes extracted from the comprehensive gene list obtained from the Genecode Release 29 (GRCh37). For the noncoding elements, we use the active tissue-specific candidate cis-regulatory elements gathered from ENCODE 3[25].

### Power of detection for the significantly recurrent peaks in *BPpc*

We used a binomial model to analyze the power of detection of significant peaks, defined as the probability to find the expected number of samples for predicting the significant peaks, similar to the approach used by PCAWG[5,69]. The procedure consisted of computing the minimum number of samples needed to reach 90% probability of a significant peak using the probability that a patient will have at least one SV in a significant peak from each cancer-specific background model $p_0 = 1 − (1 − \mu f_p)^L$, where $\mu$ is the cancer-specific average SV rate per megabase, $f_p$ is the peak breakpoint rate factor, $L$ is the median length of the top 2% peaks from each cancer-specific background model. Then the signal of detection power was calculated as a function of cohort size using the alternative probability $p_1 = 1 − (1 − p_0) \times (1 − r \times s)$ for different peak sample frequencies or prevalence ($r$ = 2%, 5%, 25%) and a fixed detection sensitivity ($s$ = 90%). We calculated the detection power across the range (minimum, maximum) of the genomic length of the top 2% peaks to evaluate the influence of proximal or distal SVs (Supplementary Fig. 7).

### CSVDriver workflow

CSVDriver aims to identify cancer-driving rearrangement events by analyzing the focal trend of breakpoint clustering. The method has been implemented in R version 3.6.2 (2019-12-12), and the code is publicly available at https://github.com/khuranalab/CSVDriver https://doi.org/10.5281/zenodo.6969761[78].

**Input and preprocessing of data.** The method takes as input the data of cancer somatic SVs calls and analyzes the combined impact of all rearrangement types, including insertions (Ins), deletions (Del), head-to-head inversions (h2hInv), tail-to-tail inversions (t2tInv), and translocations (Tra). The expected file format is SV-table containing 12 columns named as: (cohort_code, donor_id, variant_type, sv_id, chr_from, chr_from_bkpt, chr_from_bkpt, chr_from_strand, chr_to, chr_to_bkpt, chr_to_bkpt, chr_to_strand). Each row defines a single SV by the cohort (cancer type), the donor ID, the SV type (Ins, Del, Inv, Tra), the SVs unique ID, and the genome coordinate [chr, start, end, strand] for both rearrangement breakpoints (from, to). CSVDriver checks SVs from all donors and decomposes this SV-table into a BP-table sorted by genomic position per chromosome.

**Establishing the observed breakpoint proximity curve (*BPpc*).** The method bases the SV analysis on the description of the genome-wide distribution of breakpoint proximity taking all breakpoint coordinates for each distinct sample in the given cohort. Thus, if a breakpoint coordinate is present in two samples, it is represented twice. Consequently, for a cohort with several samples harboring recurrent breakpoint coordinates, they will all be included in the analysis. Then we sort all breakpoints based on their coordinates and obtained an ordered list of breakpoints encompassing all samples, each one represented by $BP_i$, where $i$ is the ordered index. Next, we annotate each $BP_i$ by computing their neighbor reachability ($BPnr_i$) defined as the average distance to reach adjacent breakpoints on both 5' and 3' sides:

$$BPnr_i = (\Delta(BP_i, BP_{i-1}) + \Delta(BP_i, BP_{i+1}))/2 \qquad (2)$$

Because a higher distance implies lower proximity to pursue a peak-calling strategy, we define the breakpoint proximity (*BPp*) upon the *BPnr* values as the normalized reverse scale, applying logarithmic transformation:

$$BPp_i = -\log10(BPnr_i + 1) \qquad (3)$$

Then, we compute the breakpoint proximity curve (*BPpc*), which is a smooth curve resulting from the nonparametric local polynomial regression (LOESS) fitted to the $BPp_i$ values (Fig. 1a). This curve shows the trend of focal clustering of the breakpoints because the fitting result is weighted toward the nearest surrounding values. The span argument ($\alpha = 0.2$) controls the size of the surrounding interval. It reflects the interval as a proportion of the total breakpoints and regulates the grade of smoothness in the resulting curve. The *BPpc* represents the distribution of breakpoints, from which we can capture statistically significant regions of high or low proximity between breakpoints. We find that the chromosomes with few breakpoints (less than 100) do not allow the creation of a reliable smooth curve, and thus breakpoint proximity can not be modeled accurately. This often occurs in small chromosomes (e.g., chr21, chr22) for some cancer cohorts. This does not impact our results since such few breakpoints do not change the GAM, and no peaks are identified in these regions.

**Modeling the expected BPpc by using a generalized additive model (GAM) with tissue-specific genomic covariates.** The model is conceived to expand the linear regression analysis of genomic covariates by introducing the capacity to investigate the potential nonlinear relationships between genomic covariates and the distribution of breakpoints. Additionally, it accounts for the contribution of the covariates' interaction to the model. Thus, CSVDriver models the expected background *BPpc* using GAM, a parametric regression method, which models the *BPpc* (dependent variable) with respect to tissue-specific data of genomic covariates (predictors or independent covariates).

GAM is a flexible extension of generalized linear models (GLM). Using a GAM, we can fit a linear model, which allows us to consider either linear or nonlinear contributions of the genomic covariates to the model of *BPpc*. To fit the GAM, CSVDriver uses the R package 'mgcv' version 1.8-28[79,80]. The values of the observed *BPpc* for each cohort fit Gamma distribution (Supplementary Fig. 12). Therefore, the model assumes each expected *BPpc* to be generated from a Gamma distribution for the identity link function of the response. The model follows the equation:

$$\begin{aligned}
\widetilde{BPpc}_i =\ & \beta_0 + FS_i + LAD_i + \text{repClass}_i + \text{ChromMark}_i + s(RT_i)LAD_i + s(GC_i)LAD_i \\
& + s(TAD.\text{recurr}_i)LAD_i + s(TADB.\text{recurr}_i)LAD_i + s(\text{GeneDensity})LAD_i \\
& + \text{te}(\text{GeneDensity}_i, TAD.\text{recurr}_i)TAD\text{seg}_{\text{class}_i} \\
& + \text{te}(\text{GeneDensity}_i, RT_i)TAD\text{seg}_{\text{class}_i} \\
& + \text{te}(RT_i, TAD.\text{recurr}_i)TAD\text{seg}_{\text{class}_i}
\end{aligned}$$

$$(4)$$

where $i = 1,\dots,N$ (total number of breakpoints), $\widetilde{BPpc}_i$ is the expected breakpoints proximity, $\beta_0$ is the intercept and the remaining terms are the genomic covariates used as predictors. The covariates FS, LAD, repClass, and ChromMark (chromatin marks) are modeled as linear factorial terms. The thin plate regression spline smooth function (s) can describe the nonlinearity in the contribution of the covariates (RT, GC, GeneDensity, TAD, and TAD-B recurrence), and for each one, the model gets the interaction with the status of the factor LAD. The model investigates the main effects of the predictors as well as the effects of the tensor product interaction (te) for the interplay between GeneDensity and TAD recurrence; GeneDensity and RT; and TAD and RT. This accounts for the status of TAD-segment class. Nonetheless, a high number of interaction terms increases the chance of over-fitting. Furthermore, GAM can be computationally expensive to reach convergence. Consequently, we try to balance the complexity of the model and its ability to explain the deviance by covariate interactions reasonably by including the interaction only of covariates that contribute at least 10% in at least one cohort in single predictor models.

We checked the effect of adding the mutational status of DNA-repair genes to the model (Supplementary Data 11, extended GAM equation). This analysis was performed on the breast cancer cohort because it is representative of the high mutational impact of DNA-repair genes. The breast cancer cohort shows 20 different DNA-repair genes (Supplementary Data 11) mutated across 104 samples (49.8%). We observe that including the mutational status of the DNA-repair genes improved model performance by only 0.2%, from 18.8 to 19%. This result corroborates the idea that the mutational status of DNA-repair genes impacts the overall breakpoint proximity signal and is thus accounted for by the original model.

**Computing the adjusted BPpc.** The goal of CSVDriver is to capture the loci where breakpoint clusters potentially arise due to selective pressure unlike the clusters associated with neutral non-selective forces. Therefore, the adjusted curve $\dot{BPpc} = BPpc - \widetilde{BPpc}$ represents a corrected BPpc where regions with values close to the $\widetilde{BPpc}$ will be considered expected biases. The signal in the y-axis depicts how close the breakpoints are in a given region (Fig. 1c). The positive values represent regions where the breakpoints are closer than expected, while regions with negative values are loci where the breakpoints are sparser than expected from the GAM. The values around zero in the adjusted curve ($\dot{BPpc}$) correspond to regions of observed *BPpc* close to the expected one.

**Detecting the significantly recurrent peaks in the BPpc.** The method takes the regions corresponding to the top 25% of the summit of each peak, which represent the regions of local maximum breakpoint clustering. Then we identify the loci that potentially correspond to the positively selected regions (Fig. 1d). Each peak is described by its peak

recurrence score (*PRs*) defined in Eq. (1). Thus, *PRs* is the highest for peaks where many samples (high *Nsmp*) contribute to create tight clusters of breakpoints (high *Peak$_A$*) over narrow regions (low *Peak$_{GR}$*). The score is square root transformed to reduce the dispersion in the values while keeping the trend of interest. We take into account the inherent limitation of the background model and the differential explained deviance per tissue type (Fig. 2a). Consequently, the approach does not use any absolute pre-established threshold, instead, it considers the cohort-specific distributions of all *PRs* (Fig. 4b density distribution) as the expectation of the combined effect of several processes. Next, for each cohort, we identify peaks with *PRs* significantly higher (Fig. 4b, QQ-plots) than the fitted Gamma distribution theoretical density using the 'fitdistrplus' R Package version 1.1-1[81]. After controlling the false discovery rate (FDR)[82] for multiple hypothesis testing, the significant loci were defined as those with FDR < 0.2.

**Detecting peaks of sample-specific rearrangements in the *BPpc*.** Additionally, the model annotates significant peaks arising from a single sample or fewer samples than expected. We compute the single-sample rearrangement score (*SSRs*) as the proportion of breakpoints per sample for each peak:

$$SSRs = N_{BP}/N_{smp} \qquad (5)$$

where $N_{BP}$ is the number of breakpoints and $N_{smp}$ is the number of samples within each peak. The *SSRs* evaluates the average contribution of each sample to the total number of breakpoints within each peak. The distribution of the peaks' *SSRs* reflects the expected empirical null background distribution from which the loci with significantly high *SSRs* are detected (FDR < 0.2). These significantly higher *SSRs* peaks, in addition to the peaks arising from only one or two samples, point to breakpoint clustering regions significantly enriched in fewer samples than expected.

**Detecting the driver candidates within the significantly recurrent peaks.** The potential functional effect of the significantly rearranged regions for each cancer is directly associated with the effects on coding and noncoding elements by their deletion, disruption, or relocation. CSVDriver determines the functional elements that are the most likely targets of positive selection within the predicted significantly recurrent peak.

For each functional element (i.e., protein-coding exons, lncRNA, enhancer, CTCF-insulator) located at a significantly recurrent peak, the method computes the element SV rearrangement score (*ERS$_{SV}$*)

$$ERS_{SV} = \frac{Nsmp_E}{Nsmp} \times \frac{Nsv_E}{L_E} \qquad (6)$$

where $Nsmp_E$ is the number of unique samples that have SVs overlapping the element, $Nsmp$ is the number of unique samples in the peak, $Nsv_E$ is the number of SVs that overlap the entire element, and $L_E$ is the length of the element. High $Nsv_E$ values could be due to multiple alleles affected or multiple clonal/subclonal cell populations with SVs at the given locus. Normalization by $L_E$ accounts for longer elements that are more likely to have a larger number SVs at random. The *ERS$_{SV}$* is a metric of the relative selective pressure acting on the elements within the significantly rearranged region, and it considers the recurrence in samples and the relative number of SVs that impact the element. The functional elements with the highest *ERS$_{SV}$* are the most likely targets of positive selection within the peak. The distribution of gene lengths in the human genome ranges from less than one kilobase to several megabases[83]. While we compute *ERS$_{SV}$* for coding genes similar to the one for other elements to account for SVs that can change the entire genic region via amplification, deletion, or locus

relocation, we find that the longer genes may be broken at the genic region leading to disruption of the coding sequence via gene fusion or translocation. Hence, for coding genes within the significant peaks, we compute a second element breakpoint rearrangement score (*ERS$_{BP}$*):

$$ERS_{BP} = \frac{Nsmp_E}{Nsmp} \times \frac{Nbp_E}{L_E} \qquad (7)$$

where $Nsmp_E$ is the number of unique samples that have SVs overlapping the element, $Nsmp$ is the number of unique samples in the peak, $Nbp_E$ is the number of breakpoints that fall within the gene body, and $L_E$ is the length of the element. Normalization by $L_E$ accounts for longer elements that are more likely to have a larger number of breakpoints at random. CSVDriver reports both *ERS$_{SV}$* and *ERS$_{BP}$* for coding genes, and the one with the higher value is used to identify putative drivers.

The method reports the element rearrangement scores for the highest scoring elements of all types (protein-coding exons, enhancers, or lncRNA) at a given peak (Supplementary Fig. 6 and Supplementary Data 6). While at a specific locus altered by SVs, it is fair to assume that affected coding exons of genes are more likely to have a greater impact than the noncoding elements, CSVDriver allows the analysis of the potential importance of all possible driver elements. One peak may contain multiple driver elements, which can represent alternative paths of disrupted regulation, or even some subgroup of samples with slightly different, yet related genomic drivers. In the current analysis, if a peak contains multiple elements of different types, all elements with the highest rearrangement scores are reported in Supplementary Fig. 6 and Supplementary Data 6. However, if one of those elements is or is associated with a known cancer gene, we consider it as the most likely candidate in a given peak shown in Fig. 5.

**Results report.** For each input cohort, CSVDriver reports the graphics of the *BPpc* annotated with the driver candidates (coding genes and noncoding elements) at each significantly recurrent peak (Supplementary Fig. 6). It also provides the summary tables with the catalog of drivers (Supplementary Data 7). The tool also provides the set of plots resulting from GAM covariates modeling to analyze the nonlinearity of relationships between the genomic features and the genomic distribution of breakpoints (Supplementary Fig. 3)

### Verification of the driver candidates
We verified our results of driver candidates using CancerMine[50] and COSMIC[51]. In addition, for the significantly rearranged peaks that have enough samples with RNA-seq, we check for the significant differential expression between samples with SVs at the given locus relative to the samples without SVs.

### Reporting summary
Further information on research design is available in the Nature Research Reporting Summary linked to this article.

## Data availability
All the data generated in this study are provided in the Supplementary Information/Source Data file. The input SV data used in the study are described in Supplementary Data 1 and are available for download at https://dcc.icgc.org/releases/PCAWG. In accordance with the data access policies of the ICGC and TCGA projects, accessing potentially identifying information needs authorization granted by applying to the TCGA Data Access Committee (DAC) via dbGaP (https://dbgap. ncbi.nlm.nih.gov), and to the ICGC Data Access Compliance Office (DACO; http://icgc.org/daco). The SVs for metastatic prostate cancer are available in the supplementary material from Quigley, D.A., et al[19]. and Viswanathan, S.R., et al.[20]. The covariates input data are provided at the CSVDriver's GitHub repository (https://github.com/khuranalab/

CSVDriver/releases/tag/v0.1.0-beta.1) and the full description is provided in Supplementary Data 2. This dataset includes the unrestricted downloadable data of the tissue-specific chromatin state marks available at Roadmap Epigenomics Mapping Consortium portal (https://egg2.wustl.edu/roadmap/web_portal/), the RT datasets available at ENCODE portal (https://www.encodeproject.org/), the FS annotation available at the HumCFS database (https://webs.iiitd.edu.in/raghava/humcfs/), the repClass data from repeat masker available at the UCSC genome browser (https://hgdownload.soe.ucsc.edu/), the TADs annotations available at the 3D Genome Browser (http://3dgenome.fsm.northwestern.edu/index.html), and the LADs annotation available in the supplementary material at Akdemir, K. C. et al. 2020[16]. Cancer-Mine database (http://bionlp.bcgsc.ca/cancermine/) and COSMIC portal (https://cancer.sanger.ac.uk/cosmic) were used for verifying cancer-related genes. Source data are provided with this paper.

## Code availability

The code for CSVDriver method is available at https://github.com/khuranalab/CSVDriver (https://doi.org/10.5281/zenodo.6969761)[78]. The method uses CRAN and Bioconductor R package including: 'mgcv' version 1.8-28 for fitting the GAM, 'fitdistrplus' version 1.1-1 for fitting the Gamma distribution theoretical density, 'biovizBase' version 1.44.0 function 'GCcontent' for computing the GC content and 'kpPlotDensity' version 1.10.0 function kpPlotDensity for computing the gene density.

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

## Acknowledgements

This work is supported by NIH grant R01CA218668 to E.K., who also thanks the Irma T. Hirschl Trust and WorldQuant Foundation for their support.

## Author contributions

E.K. and A.M-F. conceived and designed the project. A.M-F. implemented the method and performed the data analysis. A.D. performed the RNA-seq differential expression analysis. A.M-F. and E.K. wrote the manuscript. E.K. supervised the project.

## Competing interests

The authors declare no competing interests.

## Additional information

**Supplementary information** The online version contains

supplementary material available at

Alexander Martinez-Fundichely or Ekta Khurana.

**Peer review information** *Nature Communications* thanks the other
anonymous reviewer(s) for their contribution to the peer review of this
work. Peer review reports are available.

**Publisher's note** Springer Nature remains neutral with regard to
jurisdictional claims in published maps and institutional affiliations.

