## [Peer review file · Nature Communications]

REVIEWER COMMENTS

Reviewer #1 (Remarks to the Author): Expert in structural variants, bioinformatics, and cancer genomics

Martinez-Fundichely et. al present a study where they examine the breakpoints of SVs in WGS cancer data, with the aim to detect breakpoints, and associated loci under positive select within a tumor. To achieve this breakpoints were analyzed with a multi variate model using tissue specific factors. This model is then released as software via github project Cancer Structural Variation Drivers (CSVDriver). It is positive to see a study which is not based on arbitrary fixed window sizes.

After reading the manuscript I have the following points.

Major:

1. From my understanding the expected breakpoints are modelled on the background observed breakpoints per system, and then corrected by observed vs the expected. How does this then affect the analysis of recurrent breakpoints? Based on this explanation it seems that recurrent breakpoints would then be removed.
2. To a certain level there seems to be cyclic reasoning in the study design. The method to model breakpoints is both developed and applied on the same dataset, the manuscript would be improved by replicating the findings in an independent dataset.
3. Across the different cancer 'types' the authors observe large variability from 10 - 50% explained variance. This warrants an more in depth discussion including potential explanations, for example but not limited to, does differences in cell type heterogeneity in different organ systems affect the analysis?
4. Following on from point 3, there is a marked difference in the number of observed peaks per cancer type. Given that the analysis already takes into account the background distribution of SVs per cancer type. What are potential explanations of this? And associated to this it's a little surprising that no signal was found in the bone cancer subgroup.
5. The method seems focused on SV breakpoints, yet breakpoints are not independent as they multiple breakpoints are linked back to an underlying structural variant. How does this lack of independence affect the analysis? And do the authors see SV type heterogeneity in the breakpoint clusters?

6. The authors appear to cluster all breakpoints found across all patients. There are different underlying causes as to why a breakpoint cluster may occur. For example an individual patient with a single complex SV event could result in a cluster, or multiple patients with SV breakpoint in the same loci could also result in a cluster. Further explanation of the clusters found, and the content of the clusters would improve the study, including potential interaction between these two concepts of clusters, from an individual, as opposed to clusters across multiple individuals.

Minor:

1. The text of the introduction should be aligned with the abstract. Is the main purpose of the study to identify SVs under positive selection, or to examine SV breakpoints? This is currently unclear after reading first the abstract and then the introduction.

2. The study uses 2 key concepts breakpoint density and breakpoint proximity, for readability of the manuscript these concepts should be clearly defined, and then consistently used. For example the breakpoint proximity curve (BPpc) is based on 'breakpoint neighbor reachability' is used to call peak summits that show relatively higher breakpoint clustering. In addition breakpoint neighbor reachability appears to be another term for distance.

3. It is unclear which breakpoints are used in the training of the model. The authors state "all SV breakpoints are considered", but does this mean all somatic SV breakpoints, or all high confidence breakpoints?

4. The tool CSVDriver appears to calculate 2 different scores, however only one is discussed in the manuscript. It is unclear which one is being used and why.

5. The tool CSVDriver requires a large amount of user input, including tissue specific resources, this could make it difficult to use. The documentation should be clear how these resources were obtained for this study, or provide example files.

6. The term 'unique sv' should be defined there are multiple interpretations of this concept and it is currently ambiguous.

7. During the analysis some chromosomes with few SVs without clear clustering were excluded, further explanation of this is required so the reader can understand why this was done and the potential impact.

8. The number of cancer types displayed across the figures is inconsistent. Whilst the intention is to provide examples, this is not always clear to the reader, and can be confusing.

Reviewer #2 (Remarks to the Author): Expert in topologically associating domains, cancer genomics, and structural variants

Martinez-Fundichely et al. present an interesting manuscript describing a novel model they have developed to investigate structural variant breakpoint occurrence from cancer whole genome sequencing data. The method accounts for a variety of genome covariates in a potentially tissue specific manner across 32 cancer types and can explain ~57% of the variance of SV distribution. They find that 3D genome features, such as TAD boundaries, are significant contributors to the model. They also use the model to nominate potential drivers, largely focusing on instances of known cancer relevant genes potentially being found in novel cancer types. Overall, I find that the manuscript is clear and well written. I think that there has been a lot of effort to model SNV distribution in the genome, but comparatively little effort to model SV distribution, so I think that this kind of approach will be welcomed. There was some effort in this area by the PCAWG structural variant group, but I think that this study nicely builds on and expands that prior work. I do think that current manuscript would benefit from a slightly more in-depth investigation of some of the predictions of the model, and I have a few suggestions below, as the manuscript currently feels a little light in terms of the depth of analysis of the predictions of the model. That being said, I believe that with modest revisions this work would be suitable for publication in Nature Communications. I have divided my comments below into major and minor comments.

Major comments:

-I would be very curious if they can use their model to estimate the power of detecting recurrent structural variant events in the genome. They identify 79 events as recurrent, and I can't help but think that this is kind of low. I would really love it if they could do some kind of simulation of structural variants with variable frequency of occurrence, and then ask how well could such events be detected from the current set of available whole genome SV data (the ~2800 patient heres). Can they detect low frequency recurrent events, say that occur in something like 1-2% of cases? Does it matter if such events are proximal (thereby more likely interacting in the model I suspect) versus long distances away (or on different chromosomes?). How many patient genomes would we need to be able to detect low

frequency recurrent SVs, or from the currently available set of genomes do we already have all the answers?

-The authors state that the variability in predictions is from 6% in lung cancer to 50% in lymph nodes. How much of this is due to the availability of relevant modelling data for each cell type (i.e., TAD boundary calls or RT in lung data)? Does this also potentially tell us that some tumors have “missing” covariates, or that they are instead more “SV driven” tumors than other types?

-I would also be very curious how much mutations DNA repair genes contribute to the predictions. The PCAWG SV group had some nice analysis of structural variant “signatures” associated with different frequencies of DNA repair genes. I would be really curious to know if either 1) they could account for such mutations in their model or 2) perhaps more simply if the occurrence of specific mutations in DNA repair genes may account for some of the tumor types where the model performs more poorly.

Minor comments:

-The model includes known fragile sites, but they also state (lines 192-193) that fragile sites come up as significant hits in their analysis of potential enriched/driver loci. I can understand why this might happen for genes like CDKN2A, but for others (FHIT, WWOX) my (perhaps uninformed) understanding was that these were not true drivers but were found in very late replicating regions of the genome. I’m curious if the authors think that this means these function in some tissue specific instances as driver events.

-I think they should give more of an explanation for what is meant by TAD recurrence in the main text, it isn’t apparent from what is written what this really means, but from the figures and discussion it appears to be an important feature.

Small Typos:

Line 148: “tissue-sepcific“

REVIEWER COMMENTS

Reviewer #1 (Remarks to the Author): Expert in structural variants, bioinformatics, and cancer genomics

Martinez-Fundichely et. al present a study where they examine the breakpoints of SVs in WGS cancer data, with the aim to detect breakpoints, and associated loci under positive select within a tumor. To achieve this breakpoints were analyzed with a multi variate model using tissue specific factors. This model is then released as software via github project Cancer Structural Variation Drivers (CSVDriver). It is positive to see a study which is not based on arbitrary fixed window sizes. After reading the manuscript I have the following points.

Author Response: We thank the reviewer for appreciating our work.

Major comments:

Major Comment 1. From my understanding the expected breakpoints are modelled on the background observed breakpoints per system, and then corrected by observed vs the expected. How does this then affect the analysis of recurrent breakpoints? Based on this explanation it seems that recurrent breakpoints would then be removed.

Author Response:

We thank the reviewer for this question and we have clarified this in the revised manuscript. First we want to clarify that recurrent breakpoints (exact same breakpoints in multiple samples) are a subset of recurrent SVs (same region shows SVs in multiple samples though the precise coordinates of breakpoints may be different in different samples). Since same (recurrent) breakpoints from distinct samples are considered as multiple breakpoints to obtain the breakpoint proximity curve, they are not removed in the analysis. Indeed, they are expected to give rise to peaks in the proximity curve. If these peaks are associated with some co-variates throughout the genome, that will be captured by the GAM as the expected trend. However, if the peaks are not explained by the co-variates, the model will identify them as candidate regions under positive selection. In fact, this is the intended goal of the model to identify recurrent SVs and breakpoints that truly correspond to positively selected regions and are not the result of background co-variates. In the list of final significant peaks in this paper (Supplementary Table 4), while all regions represent recurrent SVs (column G), 15 candidates also exhibit recurrent breakpoints (marked with an asterisk in column F).

Excerpt From Revised Manuscript:

1. Added under Methods section under ‘Establishing the observed breakpoint proximity-curve (*BPpc*)’:
... Consequently, for a cohort with several samples harboring recurrent breakpoint coordinates, they will all be included in the analysis...
2. Supplementary Table 4 has been updated, indicating the significant peaks with recurrent breakpoints (marked with an asterisk in column F).

Major Comment 2. To a certain level there seems to be cyclic reasoning in the study design. The method to model breakpoints is both developed and applied on the same dataset, the manuscript would be improved by replicating the findings in an independent dataset.

Author Response:

We thank the reviewer for the suggestion. The implementation of CSVDriver does not aim to be a predictive model for genome-wide breakpoint location. Instead, our objective is to identify regions where the breakpoint proximity is not explained by the underlying co-variates, thus, they may correspond to positive selection. However, we agree with the reviewer that the manuscript would be improved by using an independent dataset since we expect that the fit to a particular cancer type should perform reasonably well for another independent cohort of the same cancer type. To test this hypothesis, we used structural variation calls from three different ICGC cohorts including breast cancer (549 samples), prostate cancer (396 samples), and skin cancer (206 samples), each of which are independent from the PCAWG datasets used to develop the method (breast cancer (209 samples), prostate cancer (205 samples), and skin cancer (106 samples)). We compared the top significant peaks (p-value < 0.025) output by adjusting the *BPpc* with the new computed ICGC model (Supplementary Table 10a left), with the top significant peaks (p-value < 0.025) output by the previously computed PCAWG model on cohorts of the same cancer type (Supplementary Table 10a right). We confirmed that significant cancer-specific loci target the same genomic regions, regardless of the model used. Moreover, we find that all candidates identified in PCAWG cohorts are identified in the ICGC cohorts, while three additional candidates are identified in the ICGC cohorts likely due to the larger sample sizes.

Excerpt From Revised Manuscript:

- Added under Result section under ‘CSVDriver: Computational tool to identify SV drivers from whole-genome sequences’:

To test whether the model can be applied in independent cohorts of the same cancer types, we evaluated three datasets from ICGC cohorts including breast cancer (549 samples), prostate cancer (396 samples), and skin cancer (206 samples), each of which are independent from the PCAWG datasets used to develop the method. We compared the top significant peaks (p-value < 0.025) obtained by adjusting the *BPpc* with the new computed ICGC model (Supplementary Table 10a left), with the top significant peaks (p-value < 0.025) obtained using the previously computed PCAWG model on cohorts of the same cancer type (Supplementary Table 10a right). We confirmed that significant cancer-specific loci target the same genomic regions regardless of the model used. Moreover, we find that all candidates identified in PCAWG cohorts are identified in the ICGC cohorts, while three additional candidates are identified in the ICGC cohorts likely due to the larger sample sizes.

Major Comment 3. Across the different cancer 'types' the authors observe large variability from 10 - 50% explained variance. This warrants an more in depth discussion including potential explanations, for example but not limited to, does differences in cell type heterogeneity in different organ systems affect the analysis?

Author Response:

We appreciate the reviewer’s suggestion for enhancing our manuscript with an in-depth discussion on the variability of the method's performance across the different cancer types. The model's explained deviance of the breakpoint proximity distribution is affected by the capacity of the chosen set of covariates to capture the patterns of breakpoints and explain the *BPpc* distribution related to background rearrangement events (i.e., not stochastic events). As also discussed under Reviewer 2 Comment 2, either “missing co-variates” or “missing values of co-variates” for certain cancer types could potentially explain the difference in explained deviance. We find that missing values of co-variates is likely not the reason for the different explained deviance in different cancer types (Supplementary Table 2). As for the missing co-variates, as suggested by the reviewer, we checked if the cell type heterogeneity [1] is related to the explained deviance. Indeed, we find a positive correlation (Spearman correlation coefficient = 0.4) between cell type heterogeneity and explained deviance, although it is not statistically significant likely due to the low number of cancer types (n=12) (Supplementary Table 3 and Supplementary Figure 2a). Similarly, we do not find a statistically significant correlation between average number of SVs per donor and explained deviance

although the correlation coefficients is -0.1 (Supplementary Figure 2b). Thus, while we can not check the relationship of multiple features to the explained deviance due to the small number of cancer types, it is likely that signatures of structural variation [2], the evolutionary history and the clonal status of tumors [3, 4] may also play a role in the observed variability of the explained deviance for each cohort.

We added the answer to this comment in the revised manuscript.

Excerpt From Revised Manuscript:

1- Added under Result section under ‘Expected *BPpc* using genomic covariates in a GAM’:

... The difference in explained deviance between cancer types could be due to either “missing co-variates” or “missing values of co-variates” for certain cohorts. The missing values of co-variates is likely not the reason for the different explained deviance in different cancer types (Supplementary Table 2). We checked if the cell type heterogeneity [1] is related to the explained deviance and we found a positive correlation (Spearman correlation coefficient = 0.4) though it is not statistically significant likely due to the low number of cancer types (n=12) (Supplementary Table 3 and Supplementary Fig. 2a). Similarly, we do not find a statistically significant correlation between average number of SVs per donor and explained deviance although the correlation coefficient is -0.1 (Supplementary Fig. 2b). Thus, while we can not check the relationship of multiple features to the explained deviance due to the small number of cancer types, it is likely that signatures of structural variation [2], the evolutionary history and the clonal status of tumors [3, 4] may also play a role in the observed variability of the explained deviance for each cohort ...

2- Added under Discussion section:

The model's explained deviance is affected by the capacity of the chosen set of covariates to capture the patterns of breakpoints and explain the *BPpc* distribution related to background rearrangement events.

Major Comment 4. Following on from point 3, there is a marked difference in the number of observed peaks per cancer type. Given that the analysis already takes into account the background distribution of SVs per cancer type. What are potential explanations of this? And associated to this it's a little surprising that no signal was found in the bone cancer subgroup.

Author Response:

We would first like to clarify that the total number of observed peaks is different than the number of significant peaks, which constitute the final list of candidates of rearranged regions under selection. Since observed peaks are directly derived from breakpoint proximity (and later corrected with expected peak height), we expect that cohorts with more SVs per sample would have more observed total peaks and this is consistent with our results (Supplementary Figure 2c). To identify the peaks that potentially correspond to positively selected loci across the *BPpc* rearrangement landscape, we computed the peak recurrence score (*PRs*). The significantly recurrent peaks are the loci of *PRs* significantly higher than their corresponding null background distribution (Supplementary Figure 5). Bone cohort exhibits a large number of observed peaks consistent with the high number of average SVs per donor (Supplementary Figure 2c). However, it does not show any significant peaks. In fact, we find that there is a negative correlation between total number of peaks and number of significant peaks and cohorts with relatively high number of observed peaks tend to have low number of significant peaks (Supplementary Figure 2d). This indicates that cohorts with high number of SVs in the background exhibit fewer regions showing positive selection due to SVs. Moreover, we performed power calculations in response to Reviewer 2 Major Comment 1, which also provide a potential explanation for the different number of significant peaks across cohorts (Supplementary Figure 8). These calculations show that, unlike for other cancer types, the currently available cohort size for bone cancer does not provide sufficient power to detect events under positive selection – even those with high prevalence of 25% or more (Supplementary Figure 7).

We added the answer to this comment in the revised manuscript.

Excerpt From Revised Manuscript:

1- Included under Results under ‘Detecting significantly recurrent peaks in BPpc’:

... We find that cohorts with more SVs per sample have more observed total peaks as expected (Supplementary Figure 2c) but there is a negative correlation between the total number of peaks and number of significant peaks (Supplementary Figure 2d)...

2- Added a new section ‘Power of detection for the significantly recurrent peaks in BPpc’. Included in the section:

... The results of the power calculations show that although the cohort size for esophagus, stomach, lung, uterus, and breast cancers is borderline, the sample size currently available for most cohorts provides 90% power of detection for peaks with a prevalence of 25% or more (Supplementary Figure 7). However, the 94 donors in bone cancer provide ~38% detection power for such peaks... These calculations provide a potential explanation for the different number of significant peaks across cohorts

Major Comment 5. The method seems focused on SV breakpoints, yet breakpoints are not independent as they multiple breakpoints are linked back to an underlying structural variant. How does this lack of independence affect the analysis? And do the authors see SV type heterogeneity in the breakpoint clusters?

Author Response:

Our study assumes that different SV types may impact the same genomic region resulting in the same functional impact. For example, two regions of the genome may come together due to the deletion of sequence between them or intrachromosomal translocation, as observed in the case of *TMPRSS2-ERG* fusion in prostate cancer. Therefore, our method focuses on breakpoint clustering to identify significantly rearranged regions. We investigated the SV type composition of each significant peak for each cohort (Supplementary Table 6) and observed heterogeneity in the SV types within the peaks, reinforcing that a single driver candidate can be affected by different SV types and mechanisms. Nevertheless, we also found regions significantly enriched in one particular SV type compared with the fraction of SVs in the region (Supplementary Table 6). The lack of independence between the pairs of SV breakpoints does not negatively affect our analysis. Using our approach, genomic regions that harbor several focal SV breakpoints nearby might reveal significantly rearranged loci, regardless of contribution from different SV types.

Excerpt From Revised Manuscript:

- Added under Result section under ‘Putative driver candidates identified at significantly recurrent peaks in BPpc’:

We further investigated the SV type composition of each significant peak for each cohort (Supplementary Table 6). We observed heterogeneity in the SV types within the peaks, reinforcing that a single driver candidate can be affected by different SV types and mechanisms. Nevertheless, we also found regions significantly enriched in one particular SV type compared with the fraction of SV types in the region (Supplementary Table 6).

Major Comment 6. The authors appear to cluster all breakpoints found across all patients. There are different underlying causes as to why a breakpoint cluster may occur. For example an individual patient with a single complex SV event could result in a cluster, or multiple patients with SV breakpoint in the same loci could also result in a cluster. Further explanation of the clusters found, and the content of the clusters would improve the study, including potential interaction between these two concepts of clusters, from an individual, as opposed to clusters across multiple individuals.

Author Response:

We appreciate and agree with the reviewer's comment about the different underlying causes for the observed breakpoint clustering. Indeed, besides the main goal of detecting the significantly recurrent loci potentially under positive selection, the method can annotate peaks that originate from a single sample with statistical significance (Supplementary Table 9).

In the revised manuscript, we compute single sample rearrangement score $SSRs = N_{BP}/N_{smp}$, where N_{BP} is the number of breakpoints and N_{smp} is the number of samples within each peak. The $SSRs$ evaluates the average contribution of each sample to the total number of breakpoints within each peak. Higher $SSRs$ point to regions originating from fewer samples. The distribution of the peaks' $SSRs$ reflects the expected empirical null background distribution from which the loci with significantly high $SSRs$ are detected ($FDR < 0.2$). These significant peaks in addition with the peaks arising from only one or two samples lead to identification of 12 regions for which the breakpoint load is significantly enriched in less samples than expected (Supplementary Table 9). We further compared the features of the significant single-sample peaks with the multiple-sample peaks predicted to be under positive selection. We do not find any significant difference in number of SVs, peak area, peak heights, peak genomic range or distance to nearest neighboring peak (Supplementary Figure 8). We find that single-sample peaks harbor significantly less breakpoints than multiple-sample peaks as one might expect.

Excerpt From Revised Manuscript:

3- Added under Result section under 'Detecting single-sample peaks in $BPpc$ ':

Besides the main goal of detecting the significant recurrent loci potentially under positive selection, the method annotates the peaks that originate from one or small number of samples. We compute the single sample rearrangement score $SSRs$, which evaluates the average contribution of each sample to the total number of breakpoints within each peak (Method). Higher $SSRs$ point to regions originating from fewer samples. We found 12 regions for which the breakpoint load is significantly enriched in less samples than expected in addition with the peaks arising from only one or two samples (Supplementary Table 9). We further compared the features of the significant single-sample peaks with the significantly recurrent peaks predicted to be under positive selection. We do not find any significant difference in number of SVs, peak area, peak heights, peak genomic range or distance to nearest neighboring peak (Supplementary Figure 8). We find that single-sample peaks harbor significantly less breakpoints than multiple-sample peaks as one might expect.

4- Added under Method section under 'Detecting peaks of sample-specific rearrangement in the $BPpc$ ':

Additionally, the model annotates significant peaks arising from a single sample or fewer samples than expected. We compute the single sample rearrangement score ($SSRs$) as the proportion of breakpoints per sample for each peak:

$$SSRs = N_{BP}/N_{smp}$$

where N_{BP} is the number of breakpoints and N_{smp} is the number of samples within each peak. The $SSRs$ evaluates the average contribution of each sample to the total number of breakpoints within each peak. The distribution of the peaks' $SSRs$ reflects the expected empirical null background distribution from which the loci with significantly high $SSRs$ are detected ($FDR < 0.2$). These significantly higher $SSRs$ peaks in addition with the peaks arising from only one or two samples point to breakpoint clustering regions significantly enriched in less samples than expected.

Minor Comments from Reviewer 1:

Minor Comment 1. The text of the introduction should be aligned with the abstract. Is the main purpose of the study to identify SVs under positive selection, or the examine SV breakpoints? This is currently unclear after reading first the abstract and then the introduction.

Author Response:

We appreciate this reviewer's comment. We have updated the text in the Abstract.

Excerpt From Revised Manuscript:

- Changed from:

‘However, little is known about the genomic features related to the breakpoint distribution of SVs in different cancers, a prerequisite to distinguish loci under positive selection from those with neutral evolution.’

to:

‘However, identification of SVs under positive selection is a challenging task because little is known about the genomic features related to the background breakpoint distribution in different cancers.’

Minor Comment 2. The study is uses 2 key concepts breakpoint density and breakpoint proximity, for readability of the manuscript these concepts should be clearly defined, and then consistently used. For example the breakpoint proximity curve (BPpc) is based on 'breakpoint neighbor reachability' is used to call peak summits that show relatively higher breakpoint clustering. In addition breakpoint neighbor reachability appears to be another term for distance.

Author Response:

We appreciate this reviewer's comment and we have updated the revised manuscript clearly defining each term in the method section ‘Establishing the observed breakpoint proximity-curve (BPpc)’.

‘Breakpoint neighbor reachability’ ($BPnr$) was calculated for each breakpoint as the average distance to reach adjacent breakpoints on both 5’ and 3’ sides.

$$BPnr_i = (\Delta(BP_i, BP_{i-1}) + \Delta(BP_i, BP_{i+1}))/2$$

We define the breakpoint proximity (BPp) using the $BPnr$ as the normalized reverse scale, applying logarithmic transformation:

$$BPp_i = -\log_{10}(BPnr_i + 1)$$

Then, we compute the breakpoint proximity curve ($BPpc$), which is a smooth curve resulting from the nonparametric local polynomial regression (LOESS) fitted to the BPp_i values (Fig. 1a).

Minor Comment 3. It is unclear which breakpoints are used in the training of the model. The authors state "all SV breakpoints are considered", but does this mean all somatic SV breakpoints, or all high confidence breakpoints?

Author Response:

For each cancer type we use a high confidence data set of somatic SV breakpoints. It was obtained from the consensus SVs call of PCAWG structural variation Working Group 6 [2, 5].

We have updated the revised manuscript with this clarification in the method section ‘Cancer somatic structural variations data’.

Minor Comment 4. The tool CSVDriver appears to calculate 2 different scores, however only one is discussed in the manuscript. Its unclear which one is being used and why.

Author Response:

CSVDriver calculates multiple scores: peak recurrence score, element SV rearrangement score and element breakpoint rearrangement score. We think the reviewer is asking about the two different element scores, which we have explained further in the manuscript. For clarity, we describe all the three scores below:

1) The peak recurrence score (PRs) used to identify the peaks that potentially correspond to positively selected loci across the $BPpc$ rearrangement landscape.

$$PRs = \frac{N_{smp}}{N_{sv}} \times \frac{Peak_A}{Peak_{GR}}$$

where N_{smp} is the number of unique samples in the peak, N_{sv} is the number of unique SVs, $Peak_A$ is the area under the peak and $Peak_{GR}$ is the genome range of the peak.

2) The element SV rearrangement scores (ERS_{SV}) used to detect the functional elements that are the most likely targets of positive selection within the predicted significantly recurrent peaks. For each functional element (i.e. protein-coding exons, lncRNA, enhancer, CTCF-insulator) located at a significantly rearranged peak, the ERS_{SV} is computed as:

$$ERS_{SV} = \frac{N_{smp_E}}{N_{smp}} \times \frac{N_{sv_E}}{L_E},$$

where N_{smp_E} is the number of unique samples that have SVs overlapping the element, N_{smp} is the number of unique samples in the peak, N_{sv_E} is the number of SVs that overlap the entire element, and L_E is the length of the element. Normalization by L_E accounts for longer elements that are more likely to have a larger number of SVs at random.

3) The distribution of gene lengths in the human genome ranges from less than one kilobase to several megabases [6]. While we compute ERS_{SV} for coding genes similar to the one for other elements to account for SVs that can change the entire genic region via amplification, deletion, or locus relocation, we find that the longer genes may be broken at the genic region leading to disruption of the coding sequence via gene fusion or translocation. Hence, for coding genes within the significant peaks, we compute a second element breakpoint rearrangement score (ERS_{BP}):

$$ERS_{BP} = \frac{N_{smp_E}}{N_{smp}} \times \frac{N_{bp_E}}{L_E},$$

where N_{smp_E} is the number of unique samples that have SVs overlapping the element, N_{smp} is the number of unique samples in the peak, N_{bp_E} is the number of breakpoints that fall within the gene body, and L_E is the length of the element. Normalization by L_E accounts for longer elements that are more likely to have a larger number of breakpoints at random.

CSVDriver reports both ERS_{SV} and ERS_{BP} for coding genes and the one with the higher value is used to identify putative drivers.

We have updated the revised manuscript adding and clarifying the equations for the two scores.

Excerpt From Revised Manuscript:

- Added under Methods section under ‘Detecting the driver candidates within the significantly recurrent peaks:

For each functional element (i.e. protein-coding exons, lncRNA, enhancer, CTCF-insulator) located at a significantly recurrent peak, the method computes the element SV rearrangement score (ERS_{SV})

$$ERS_{SV} = \frac{N_{smp_E}}{N_{smp}} \times \frac{N_{sv_E}}{L_E}$$

where $Nsmp_E$ is the number of unique samples that have SVs overlapping the element, $Nsmp$ is the number of unique samples in the peak, Nsv_E is the number of SVs that overlap the entire element and L_E is the length of the element...

...The distribution of gene lengths in the human genome ranges from less than one kilobase to several megabases [6]. While we compute ERS_{SV} for coding genes similar to the one for other elements to account for SVs that can change the entire genic region via amplification, deletion, or locus relocation, we find that the longer genes may be broken at the genic region leading to disruption of the coding sequence via gene fusion or translocation. Hence, for coding genes within the significant peaks, we compute a second element breakpoint rearrangement score (ERS_{BP}):

$$ERS_{BP} = \frac{Nsmp_E}{Nsmp} \times \frac{Nbp_E}{L_E},$$

where $Nsmp_E$ is the number of unique samples that have SVs overlapping the element, $Nsmp$ is the number of unique samples in the peak, Nbp_E is the number of breakpoints that fall within the gene body, and L_E is the length of the element. Normalization by L_E accounts for longer elements that are more likely to have a larger number of breakpoints at random. CSVDriver reports both ERS_{SV} and ERS_{BP} for coding genes and the one with the higher value is used to identify putative drivers.

Minor Comment 5. The tool CSVDriver requires a large amount of user input, including tissue specific resources, this could make it difficult to use. The documentation should be clear how these resources were obtained for this study, or provide example files.

Author Response:

We thank the reviewer for this comment. The revised Supplementary Table 2 better describes the input data for running the method, and their corresponding sources. We added information about the sources of the data for nine covariates of the model. For example, we added the cell lines from which replication timing was obtained, the source for LAD annotation, the information for the TAD datasets and the ChromHMM annotations for chromatin marks. In addition, we also provide documentation with the code at the GitHub repository <https://github.com/khuranalab/CSVDriver/blob/main/README.md> to demonstrate how to run it with example input and output files.

Minor Comment 6. The term 'unique sv' should be defined there are multiple interpretations of this concept and its currently ambiguous.

Author Response:

We used the definition of 'unique SV' as the unique combination of breakpoint coordinates, SV type and sample ID. However, we realize that indeed the word 'unique' is not needed, thus we have removed it from the text to avoid confusion.

Minor Comment 7. During the analysis some chromosomes with few SVs without clear clustering were excluded, further explanation of this is required so the reader can understand why this was done and the potential impact.

Author Response:

The $BPpc$ represents the distribution of breakpoints, from which we can capture statistically significant regions of high or low proximity between breakpoints. We find that the chromosomes with few breakpoints (less than 100) do not allow the creation of a reliable smooth curve and thus breakpoint proximity can not be modeled accurately. This

often occurs in small chromosomes (e.g., chr21, chr22) for some cancer cohorts. This does not impact our results since such few breakpoints do not change the GAM and no peaks are identified in these regions.

Excerpt From Revised Manuscript :

- Included under Methods under ‘Establishing the observed breakpoint proximity-curve (*BPpc*)’:

The *BPpc* represents the distribution of breakpoints, from which we can capture statistically significant regions of high or low proximity between breakpoints. We find that the chromosomes with few breakpoints (less than 100) do not allow the creation of a reliable smooth curve and thus breakpoint proximity can not be modeled accurately. This often occurs in small chromosomes (e.g., chr21, chr22) for some cancer cohorts. This does not impact our results since such few breakpoints do not change the GAM and no peaks are identified in these regions

Minor Comment 8. The number of cancer types displayed across the figures is inconsistent. Whilst the intention is to provide examples, this is not always clear to the reader, and can be confusing.

Author Response:

While Fig. 2, Fig. 4C and Fig. 5 show all cancer cohorts, same representative cohorts are shown in Fig. 3 and Fig. 4 a, b. We now point clearly to the Supplementary Figure number where the other cohorts are shown in the main Figure legend. Fig. 6 shows the promising candidates for which the gene is known to be oncogenic in another cancer type, and only the cancer types in which these candidates were identified are shown.

Reviewer #2 (Remarks to the Author): Expert in topologically associating domains, cancer genomics, and structural variants

Martinez-Fundichely et al. present an interesting manuscript describing a novel model they have developed to investigate structural variant breakpoint occurrence from cancer whole genome sequencing data. The method accounts for a variety of genome covariates in a potentially tissue specific manner across 32 cancer types and can explain ~57% of the variance of SV distribution. They find that 3D genome features, such as TAD boundaries, are significant contributors to the model. They also use the model to nominate potential drivers, largely focusing on instances of known cancer relevant genes potentially being found in novel cancer types. Overall, I find that the manuscript is clear and well written. I think that there has been a lot of effort to model SNV distribution in the genome, but comparatively little effort to model SV distribution, so I think that this kind of approach will be welcomed. There was some effort in this area by the PCAWG structural variant group, but I think that this study nicely builds on and expands that prior work. I do think that current manuscript would benefit from a slightly more in-depth investigation of some of the predictions of the model, and I have a few suggestions below, as the manuscript currently feels a little light in terms of the depth of analysis of the predictions of the model. That being said, I believe that with modest revisions this work would be suitable for publication in Nature Communications. I have divided my comments below into major and minor comments.

Author Response: We thank the reviewer for appreciating the manuscript and for the suggestions which have helped make the manuscript stronger.

Major comments:

Major comment 1.1. I would be very curious if they can use their model to estimate the power of detecting recurrent structural variant events in the genome. They identify 79 events as recurrent, and I can't help but think that this is kind of low.

Author Response:

We thank the reviewer for these suggestions about detection power of significant peaks. Regarding the number of significant loci predicted in our study, we would first like to clarify that our method identifies the loci which potentially correspond to regions of positive selection since their high breakpoint proximity pooled across samples can not be explained by other co-variates. Thus, while these regions correspond to recurrent SVs, not all recurrent SVs exhibit regions under positive selection, forming the basis of our method. Indeed, there are more events that are recurrent since there are 1184 peaks (average of 75 peaks per cancer type) that exhibit SVs in multiple samples but most of them likely do not represent positively selected regions and may be explained by the co-variates of breakpoint distribution (Supplementary Table 2). We would also like to point out that the PCAWG study identified 53 regions with significantly recurrent breakpoints (SRB) using a pan-cancer approach of enrichment in breakpoint density across genomic windows (50 kb bins). This number is actually lower than the number of significant events identified in our work (74 for primary cancers to compare with the PCAWG cohort and 5 additional from metastatic prostate cancer which was not included in the PCAWG cohort). In the manuscript, we indicated that 42 peaks overlap with the PCAWG regions, whereas the 32 peaks that do not overlap with the PCAWG results significantly capture cancer type-specific rearranged loci (Supplementary Table 5).

Major comment 1.2. I would really love it if they could do some kind of simulation of structural variants with variable frequency of occurrence, and then ask how well could such events be detected from the current set of available whole genome SV data (the ~2800 patient heres). Can they detect low frequency recurrent events, say that occur in something like 1-2% of cases?

Author Response:

We thank the reviewer for this excellent suggestion and we have added power calculations in the revised manuscript. We note that due to the complex relationship of genomic co-variables and SV breakpoints, simulations of SVs may not represent the null accurately. Instead, we used a binomial model to analyze the power of detection of significant peaks defined as the probability to find the expected number of samples for predicting the significant peaks, similar to the approach used for power calculations by PCAWG [7, 8]. The procedure consisted of computing the minimal number of samples needed to reach 90% probability of a significant peak using the probability that a patient will have at least one SV in a significant peak from each cancer-specific background model $p_0 = 1 - (1 - \mu f_p)^L$, where μ is the cancer-specific average SV rate per megabase, f_p is the peak breakpoint rate factor, L is the median length of the top 2% peaks from each cancer-specific background model. Then the signal of detection power was calculated as a function of cohort size using the alternative probability $p_1 = 1 - (1 - p_0) \times (1 - r \times s)$ for different peak sample frequencies or prevalence ($r = 2\%, 5\%, 25\%$) and a fixed detection sensitivity ($s = 90\%$) (Supplementary Fig. 7).

The results of the power calculations shown in Supplementary Figure 7 show that although the cohort size for esophagus, stomach, lung, uterus, and breast cancers is borderline, the sample size currently available for most cohorts provides 90% power of detection for peaks with a prevalence of 25% or more. However, the 94 donors in bone cancer provide ~38% detection power for such peaks. We found that for lower frequency events (5% or lower), the detection power ranges from ~10% to ~75%, so the sample size is insufficient for any cohort to reach 90% power of detection.

Excerpt From Revised Manuscript:

1- Included under Result under ‘Power of detection for the significantly recurrent peaks in *BPpc*’:

We used a binomial model to analyze the power of detection of significant peaks defined as the probability to find the expected number of samples for predicting the significant peaks, similar to the approach used by PCAWG [7, 8]. The results of the power calculations shown in Supplementary Figure 7 show that although the cohort size for esophagus, stomach, lung, uterus, and breast cancers is borderline, the sample size currently available for most cohorts provides 90% power of detection for peaks with a prevalence of 25% or more. However, the 94 donors in bone cancer provide ~38% detection power for such peaks. We found that for lower frequency events (5% or lower), the detection power ranges from ~10% to ~75%, so the sample size is insufficient for any cohort to reach 90% power of detection.

2- Included under Methods under ‘Power of detection for the significantly recurrent peaks in *BPpc*’:

We used a binomial model to analyze the power of detection of significant peaks defined as the probability to find the expected number of samples for predicting the significant peaks, similar to the approach used by PCAWG [7, 8]. The procedure consisted of computing the minimal number of samples needed to reach 90% probability of a significant peak using the probability that a patient will have at least one SV in a significant peak from each cancer-specific background model $p_0 = 1 - (1 - \mu f_p)^L$, where μ is the cancer-specific average SV rate per megabase, f_p is the peak breakpoint rate factor, L is the median length of the top 2% peaks from each cancer-specific background model. Then the signal of detection power was calculated as a function of cohort size using the alternative probability $p_1 = 1 - (1 - p_0) \times (1 - r \times s)$ for different peak sample frequencies or prevalence ($r = 2\%, 5\%, 25\%$) and a fixed detection sensitivity ($s = 90\%$).

Major comment 1.3. Does it matter if such events are proximal (thereby more likely interacting in the model I suspect) versus long distances away (or on different chromosomes?).

The proximal SVs are likely to give rise to narrow peaks in our method while SVs whose breakpoints are far apart in individual samples may give rise to narrow peaks (if the breakpoints cluster at a narrow region across samples) or wide peaks. We expect the power of our method can be different for narrow vs. wide peaks in the breakpoint proximity curve. Thus, we calculated the detection power across the range (minimum, maximum) of genomic length of the top 2% peaks to evaluate the influence of proximal or distal SVs. We found that smaller genomic regions (narrow peaks) need fewer samples to get 90% of the detection power while larger genomic areas (wider peaks) need more samples to reach that power (Supplementary Fig. 7). When the number of samples exceeds the sample size of the current cohorts, some cases show ambiguity in this trend for low prevalence predictions. This is likely due to the uncertainty in the empirical estimate of the breakpoint rate factor and peak's genomic length used in the binomial model which may change with increased sample size.

Excerpt From Revised Manuscript:

- 1- Included under Methods under 'Power of detection for the significantly recurrent peaks in *BPpc*':
... We calculated the detection power across the range (minimum, maximum) of genomic length of the top 2% peaks to evaluate the influence of proximal or distal SVs (Supplementary Fig. 7).
- 2- Included under Result under 'Power of detection for the significantly recurrent peaks in *BPpc*':
... We found that smaller genomic regions (narrow peaks) need fewer samples to get 90% of the detection power while larger genomic areas (wider peaks) need more samples to reach that power (Supplementary Fig. 7). When the number of samples exceeds the sample size of the current cohorts, some cases show ambiguity in this trend for low prevalence predictions. This is likely due to the uncertainty in the empirical estimate of the breakpoint rate factor and peak's genomic length used in the binomial model which may change with increased sample size.

Major comment 1.4. How many patient genomes would we need to be able to detect low frequency recurrent SVs, or from the currently available set of genomes do we already have all the answers?

The power calculations shown in Supplementary Figure 7 show that for events of frequency 5% or lower, we do not have sufficient power for any of the cohorts. While at least 100 to 200 genomes are needed for most cohorts, others including prostate and kidney need more than 250 genomes, whereas liver, brain, and pancreas need more than 300. The number of genomes available for each cohort is marked in Supplementary Figure 7.

Excerpt From Revised Manuscript:

- 1- Included under Result under 'Power of detection for the significantly recurrent peaks in *BPpc*':
The power calculations shown in Supplementary Figure 7 show that for events of frequency 5% or lower, we do not have sufficient power for any of the cohorts. While at least 100 to 200 genomes are needed for most cohorts, others including prostate and kidney need more than 250 genomes, whereas liver, brain, and pancreas need more than 300. The number of genomes available for each cohort is marked in Supplementary Figure 7.
- 2- Included under Discussion:
Finally, larger sample sizes are needed to comprehensively capture all the cancer-specific significantly rearranged regions under positive selection particularly for events with low prevalence.

Major comment 2. The authors state that the variability in predictions is from 6% in lung cancer to 50% in lymph nodes. How much of this is due to the availability of relevant modelling data for each cell type (i.e., TAD boundary calls or RT in lung data)? Does this also potentially tell us that some tumors have "missing" covariates, or that they are instead more "SV driven" tumors than other types?

Author Response:

The available information on replication timing does not cover all tissue types of interest. We used the average of Repli-seq data (1 Mb window) from the different tissues and cell lines available in ENCODE, (Supplementary Table 2). Similarly, the TAD data is not available for all tissue type in the study (Supplementary Table 2). However, the difference in the “missing values” in the covariate data availability for each tissue type does not explain the variability in the performance of the model (see below Summary from Supplementary Table 2). We found that the breakpoint proximity distribution of lung and liver cohorts is the least explained by our model, yet for these cohorts the replication timing and TAD data is available. As for “SV driven” tumors, in fact we observe that cancer types with higher explained deviance tend to have more significant peaks predicted to be under positive selection, thus the ones with lower explained deviance do not seem to be more SV driven (Supplementary Table 2). This suggests that there might indeed be “missing covariates” that lead to low explained deviance in some cancer cohorts. We have also addressed the differential explained deviance for different cancer types in response to Reviewer 1 Comment 3 and added additional text in Discussion.

Summary from Supplementary Table 2 about available data per tissue type

	TAD	RT	Chromatin mark	explained deviance	N° significant peaks
lung	yes	yes	yes	10.30%	1
liver	yes	yes	yes	11.90%	5
pancreas	yes	no	yes	12.10%	3
skin	yes	yes	yes	15.00%	1
breast	yes	yes	yes	18.80%	1
ovary	no	no	yes	23.20%	2
kidney	yes	no	yes	23.80%	2
prostate-PRI	yes	yes	yes	24.90%	6
prostate-MET	yes	yes	yes	29.40%	5
bone	yes	no	yes	29.60%	0
stomach	no	no	yes	29.90%	7
uterus	no	no	yes	34.70%	1
brain	yes	yes	yes	36.20%	11
esophagus	no	no	yes	36.20%	11
Colorectal	no	no	yes	36.50%	13
lymph-nodes	yes	yes	yes	57.10%	10

Excerpt From Revised Manuscript

1- Added under Result section under ‘Expected *BPPc* using genomic covariates in a GAM’:

... The difference in explained deviance between cancer types could be due to either “missing co-variates” or “missing values of co-variates” for certain cohorts. The missing values of co-variates is likely not the reason for the different explained deviance in different cancer types (Supplementary Table 2). We checked if the cell type heterogeneity [1] is related to the explained deviance and we found a positive correlation (Spearman correlation coefficient = 0.4) though it is not statistically significant likely due to the low number of cancer types (n=12) (Supplementary Table 3 and Supplementary Fig. 2a). Similarly, we do not find a statistically significant correlation between average number of SVs per donor and explained deviance although the correlation coefficient is -0.1 (Supplementary Fig. 2b). Thus, while we can not check the relationship of multiple features to the explained deviance due to the small number of cancer types, it is likely that signatures of structural variation [2], the evolutionary history and the clonal status of tumors [3, 4] may also play a role in the observed variability of the explained deviance for each cohort ...

2- Added under Discussion section:

...The model's explained deviance is affected by the capacity of the chosen set of covariates to capture the patterns of breakpoints and explain the *BPpc* distribution related to background rearrangement events. The variability in the model's performance across cancer types suggests that some missing cancer-specific covariates should be investigated.

Major comment 3. I would also be very curious how much mutations DNA repair genes contribute to the predictions. The PCAWG SV group had some nice analysis of structural variant "signatures" associated with different frequencies of DNA repair genes. I would be really curious to know if either 1) they could account for such mutations in their model or 2) perhaps more simply if the occurrence of specific mutations in DNA repair genes may account for some of the tumor types where the model performs more poorly.

Author Response:

We thank the reviewer for the suggestion to analyze the impact of the mutational status of DNA-repair genes on the *BPpc* signal [2]. SV-signatures active in cancer patients constitute a major cause of the total burden of SVs and breakpoint genomic distribution. Because our cancer-specific *BPpc* approach models the different patterns of breakpoint clustering based on nine different genomic covariates (Supplementary Table 2), we hypothesized that breakpoint proximity signal inherently captures the impact of the affected DNA-repair genes.

To test this hypothesis, we followed the reviewer's suggestion and compared our current results with results output from a model that accounts for mutational status of DNA-repair genes as a covariate (see extended GAM equation below). Because breast tumors are representative of the high mutational impact of DNA repair genes, this analysis was performed on the breast cancer cohort. Our breast cancer cohort shows 20 different DNA-repair genes (Supplementary Table 11) mutated across 104 samples (49.8%). If mutational status of DNA-repair genes is a missing factor in our model, we expect an increase in model performance when accounting for this variable. However, we observe that including the mutational status of the DNA repair genes improved model performance by only 0.2% from 18.8% to 19%. This result corroborates the idea that the mutational status of DNA repair genes impacts the overall breakpoint proximity signal and is thus accounted for by the original model.

Equation of the extended model including the mutational status of DNA-repair genes as covariate:

$$\begin{aligned} \widehat{BPpc}_i = & \beta_0 + FS_i + LAD_i + repClass_i + ChromMark_i + ATM_{Si} + BRCA1_{Si} + BRCA2_{Si} + CDK12_{Si} \\ & + CHEK2_{Si} + FANCA_{Si} + FANCC_{Si} + FANCD2_{Si} + FANCF_{Si} + FANCG_{Si} + FANCL_{Si} \\ & + FANCM_{Si} + MSH2_{Si} + MSH3_{Si} + MSH4_{Si} + MSH6_{Si} + PALB2_{Si} + POLE_{Si} + RAD51B_{Si} \\ & + TP53_{Si} + s(RT_i)LAD_i + s(GC_i)LAD_i + s(TAD.recurr_i)LAD_i + s(TADB.recurr_i)LAD_i \\ & + s(GeneDensity)LAD_i + te(GeneDensity_i, TAD.recurr_i)TADsegm_{class_i} \\ & + te(GeneDensity_i, RT_i)TADsegm_{class_i} + te(RT_i, TAD.recurr_i)TADsegm_{class_i} \end{aligned}$$

where we kept the nine covariates in the model as in the original CSVDriver and added the 20 new linear factorial covariate terms that account for the mutational status of each DNA-repair gene in the patient of each breakpoint *i*.

Excerpt From Revised Manuscript

- Included under Methods under 'Modeling the expected BPpc by using generalized additive model (GAM) with tissue-specific genomic covariates':

We checked the effect of adding the mutational status of DNA-repair genes to the model (Supplementary Table 11, extended GAM equation). This analysis was performed on the breast cancer cohort because it is representative of the high mutational impact of DNA repair genes. The breast cancer cohort shows 20 different DNA-repair genes (Supplementary Table 11) mutated across 104 samples (49.8%). We observe that including the mutational status of the DNA repair genes improved model performance by only 0.2% from 18.8% to 19%. This result corroborates the idea that the mutational status of DNA repair genes impacts the overall breakpoint

proximity signal and is thus accounted for by the original model.

Minor comments from Reviewer 2:

Minor comment 1. The model includes known fragile sites, but they also state (lines 192-193) that fragile sites come up as significant hits in their analysis of potential enriched/driver loci. I can understand why this might happen for genes like *CDKN2A*, but for others (*FHIT*, *WWOX*) my (perhaps uninformed) understanding was that these were not true drivers but were found in very late replicating regions of the genome. I'm curious if the authors think that this means these function in some tissue specific instances as driver events.

Author Response:

We understand the reviewer concern about the genes *FHIT* and *WWOX* because an analysis of structural signatures did not find evidence of positive selection over these fragile sites [9]. We would like to note that analysis of homozygous deletions was performed in a pan-cancer manner, and it remains possible that the previous approach missed the detection of low levels of positive selection over these genes, as also stated by the authors of that paper. In addition, there is evidence of modestly increased tumor burden at long latency in mice with *FHIT* and *WWOX* knockout [10]. While it remains unclear whether the regions of fragile sites have meaningful implications in cancer progression, by accounting for fragile sites as co-variables in our model, our findings reveal the specific cancer types where these fragile site genes are more likely to have tumorigenesis activity via increased genomic instability [11, 12].

Excerpt From Revised Manuscript

- Added under Results section under 'Putative driver candidates identified at significantly recurrent peaks in *BPpc*':
... While it remains unclear whether the regions of fragile sites have meaningful implications in cancer progression, by accounting for these sites as co-variables in our model, our findings reveal the specific cancer types where these fragile site genes are more likely to have tumorigenesis activity via increased genomic instability [11, 12] ...

Minor comment 2. I think they should give more of an explanation for what is meant by TAD recurrence in the main text, it isn't apparent from what is written what this really means, but from the figures and discussion it appears to be an important feature.

Author Response:

The recurrence of each TAD region was computed as the number of samples (cell lines and tissues) with a minimum of 70% overlap of the TAD regions divided by the total number of samples. TADs with low recurrence (< 0.5) point to the regions with high 3D structural variability across tissues and potential for tissue specificity, while TADs with high recurrence (> 0.5) are regions with similar structure across all datasets.

We have clarified the meaning of TAD recurrence in the revised manuscript.

Excerpt From Revised Manuscript:

- Included under Methods under 'Genomic feature annotations and tissue-specific epigenomic covariates':
... For our tissue-specific analysis, we use the recurrence of TAD and TAD boundaries (TAD-B) regions. The recurrence of each TAD region was computed as the number of samples (cell lines and tissues) with a minimum of 70% overlap of the TAD regions divided by the total number of samples. TADs with low recurrence (< 0.5)

point to the regions with high 3D structural variability across tissues and potential for tissue specificity, while TADs with high recurrence (> 0.5) are regions of similar structure across all datasets...

Minor comment 3. Small Typos:

Line 148: "tissue-sepcific"

Author Response:

Thanks so much, we corrected that typo.

Response references

1. Zhang, K., et al., *A single-cell atlas of chromatin accessibility in the human genome*. Cell, 2021. **184**(24): p. 5985-6001 e19.
2. Li, Y., et al., *Patterns of somatic structural variation in human cancer genomes*. Nature, 2020. **578**(7793): p. 112-121.
3. Gerstung, M., et al., *The evolutionary history of 2,658 cancers*. Nature, 2020. **578**(7793): p. 122-128.
4. Dentre, S.C., et al., *Characterizing genetic intra-tumor heterogeneity across 2,658 human cancer genomes*. Cell, 2021. **184**(8): p. 2239-2254 e39.
5. Consortium, I.T.P.-C.A.o.W.G., *Pan-cancer analysis of whole genomes*. Nature, 2020. **578**(7793): p. 82-93.
6. Lopes, I., et al., *Gene Size Matters: An Analysis of Gene Length in the Human Genome*. Front Genet, 2021. **12**: p. 559998.
7. Rheinbay, E., et al., *Analyses of non-coding somatic drivers in 2,658 cancer whole genomes*. Nature, 2020. **578**(7793): p. 102-111.
8. Rheinbay, E., et al., *Recurrent and functional regulatory mutations in breast cancer*. Nature, 2017. **547**(7661): p. 55-60.
9. Bignell, G.R., et al., *Signatures of mutation and selection in the cancer genome*. Nature, 2010. **463**(7283): p. 893-8.
10. Iliopoulos, D., et al., *Roles of FHIT and WWOX fragile genes in cancer*. Cancer Lett, 2006. **232**(1): p. 27-36.
11. Le Tallec, B., et al., *Common fragile site profiling in epithelial and erythroid cells reveals that most recurrent cancer deletions lie in fragile sites hosting large genes*. Cell Rep, 2013. **4**(3): p. 420-8.
12. Lukusa, T. and J.P. Fryns, *Human chromosome fragility*. Biochim Biophys Acta, 2008. **1779**(1): p. 3-16.

REVIEWERS' COMMENTS

Reviewer #1 (Remarks to the Author):

I have no further comments, and am satisfied how the authors have responded to my initial review.

Reviewer #3 (Remarks to the Author): Expert in TADs, 3D chromatin organisation, and cancer epigenomics

The authors have responded well to the comments of the two initial reviewers, they supported their answers with clarifications and added additional data analysis.